# Improved Coresets for Euclidean $k$-Means

**Vincent Cohen-Addad**
Google Research

**Kasper Green Larsen**
Aarhus University

**David Saulpic**
University of Vienna

**Chris Schwiegelshohn**
Aarhus University

**Omar Ali Sheikh-Omar**
Aarhus University

## Abstract

Given a set of $n$ points in $d$ dimensions, the Euclidean $k$-means (resp. Euclidean $k$-median) problem consists of finding $k$ centers such that the sum of squared distances (resp. sum of distances) from every point to its closest center is minimized. The arguably most popular way of dealing with this problem in the big data setting is to first compress the data by computing a weighted subset known as a coreset and then run any algorithm on this subset. The guarantee of the coreset is that for any candidate solution, the ratio between coreset cost and the cost of the original instance is less than a $(1 \pm \varepsilon)$ factor. The current state of the art coreset size is $\tilde{O}(\min(k^2 \cdot \varepsilon^{-2}, k \cdot \varepsilon^{-4}))$ for Euclidean $k$-means and $\tilde{O}(\min(k^2 \cdot \varepsilon^{-2}, k \cdot \varepsilon^{-3}))$ for Euclidean $k$-median. The best known lower bound for both problems is $\Omega(k\varepsilon^{-2})$. In this paper, we improve these bounds to $\tilde{O}(\min(k^{3/2} \cdot \varepsilon^{-2}, k \cdot \varepsilon^{-4}))$ for Euclidean $k$-means and $\tilde{O}(\min(k^{4/3} \cdot \varepsilon^{-2}, k \cdot \varepsilon^{-4}))$ for Euclidean $k$-median. In particular, ours is the first provable bound that breaks through the $k^2$ barrier while retaining an optimal dependency on $\varepsilon$.

## 1 Introduction

Coresets have become a staple in the design of algorithms for large data sets. In the most general setting, a coreset compresses the data set in such a way that for any set of previously specified candidate queries, the cost of evaluating the query and the cost of the coreset are similar, up to an arbitrary small distortion.

A popular subject in coreset literature is the Euclidean $k$-means problem. Here, we are given $n$ points $P$ in $d$ dimensions and our task is to find a set of $k$ points $C$ called centers minimizing $\text{cost}(P, C) := \sum_{p \in P} \min_{c \in C} \|p - c\|^2$, where $\|p - c\| = \sqrt{\sum_{i=1}^{d} (p_i - c_i)^2}$ denotes the Euclidean distance. In this case, a coreset is a weighted subset of the input such that difference between the cost for any set of $k$ centers $C$ on the coreset and the cost on the original point set $P$ is at most $\varepsilon \cdot \text{cost}(P, C)$. Since its initial study by Har-Peled and Mazumdar [2004], the Euclidean $k$-means problem has received arguably the most attention out of any coreset problem. The current state of the art by Cohen-Addad et al. [2022] yields coresets of size $\tilde{O}(k\varepsilon^{-2} \min(k, \varepsilon^{-2}))$, where $\tilde{O}(x)$ hides multiplicative factors that are polylogarithmic in $x$. Unfortunately, there is still a gap towards the best known lower bound of $\Omega(k\varepsilon^{-2})$ by Cohen-Addad et al. [2022].

We thus have the option of obtaining either an optimal dependency on $k$, at the cost of a suboptimal dependency on $\varepsilon^{-1}$, or an optimal dependency on $\varepsilon^{-1}$, at the cost of a suboptimal dependency on $k$. While these bounds suggest that the lower bound is the correct answer, things are not as clear on closer inspection. Quadratic dependencies on $k$ become necessary for many forms of analysis and so far, it is unknown how to avoid this loss while retaining an optimal dependency on the remaining

36th Conference on Neural Information Processing Systems (NeurIPS 2022).

parameters[1]. Moreover, the trade-off between the dependency on $k$ and the dependency on $\varepsilon^{-2}$ is natural. Specifically, if $k \approx \varepsilon^{-2}$ the two previous alternatives $\tilde{O}(k^2/\varepsilon^2)$ and $\tilde{O}(k/\varepsilon^4)$ are equal.

In this paper we break through this barrier. Specifically, we show that coresets of size $\tilde{O}(k^{3/2}\varepsilon^{-2})$ exist. Furthermore, for the Euclidean $k$-median problem where we minimize $\mathrm{cost}(P,C) := \sum_{p \in P} \min_{c \in C} \|p - c\|$, we also improve the known bounds from $\tilde{O}(k\varepsilon^{-2}\min(k,\varepsilon^{-1}))$ to $\tilde{O}(k\varepsilon^{-2}\min(k^{1/3},\varepsilon^{-1}))$. In our view, this is further and arguably stronger evidence that the $k\varepsilon^{-2}$ bound will be the correct answer. Another contribution exists on a technical level. Previously, most coreset constructions for high dimensions heavily relied on terminal embeddings to facilitate the analysis. In this paper, we present a novel method that avoids terminal embeddings. We expect that our technique may have further applications for coreset constructions in Euclidean spaces.

## 1.1 Techniques

Starting point of our work is the framework introduced by Cohen-Addad et al. [2021a] and specifically Cohen-Addad et al. [2022]. Prior to Cohen-Addad et al. [2022], all coreset analyses required a dependency of at least $k \cdot d \cdot \varepsilon^{-2}$. To illustrate why, suppose we are sampling points from some distribution and wish to use the sampled points as an estimator for the true cost of any candidate solution. The analysis then consists of (1) a bound on the variance $\sigma^2$ of the estimator and (2) a bound on the number of solutions $|\mathbb{N}|$ to be approximated. This typically results in coresets of size $O(\varepsilon^{-2} \cdot \sigma^2 \cdot \log|\mathbb{N}|)$. When enumerating all (discretized) candidate solutions (henceforth called a *net*) in $d$ dimensions, virtually all known techniques result in $|\mathbb{N}| \approx \exp(k \cdot d)$. The dependency on $d$ may be reduced to $\log(k/\varepsilon)\varepsilon^{-2}$ using dimension reduction techniques.

To bypass this, Cohen-Addad et al. [2022] used a chaining-based analysis to define a sequence of discretized candidate solutions. Specifically, they showed that there exist discretizations $\mathbb{N}_\alpha$ of size $\exp(k\alpha^{-2})$ such that for any point $p$ and any solution $\mathcal{S}$, there exists a solution $\mathcal{S}_\alpha$ with

$$|\mathrm{cost}(p, \mathcal{S}_\alpha) - \mathrm{cost}(p, \mathcal{S})| \leq \alpha \cdot \mathrm{cost}(p, \mathcal{S}),$$

where $\mathrm{cost}(p, \mathcal{S}_\alpha) = \min_{s \in \mathcal{S}_\alpha} \|p - s\|^2$ and $\mathrm{cost}(p, \mathcal{S}) = \min_{s \in \mathcal{S}} \|p - s\|^2$. The idea of the analysis is then to write $\mathrm{cost}(p, \mathcal{S})$ as a telescoping sum

$$\mathrm{cost}(p, \mathcal{S}) = \sum_{h=1}^{\infty} \mathrm{cost}(p, \mathcal{S}_{2^{-(h+1)}}) - \mathrm{cost}(p, \mathcal{S}_{2^{-h}})$$

and show that the sampled points achieve concentration for each summand. The number of candidate solutions is now $|\mathbb{N}_{2^{-(h+1)}}| \cdot |\mathbb{N}_{2^{-h}}| \approx \exp(k \cdot 2^{2h})$ but the difference in cost is $2^{-h}\mathrm{cost}(p, \mathcal{S})$. This directly leads to a variance of the order $2^{-2h}\sigma^2$, where $\sigma^2$ is the variance of the basic estimator. Thus, the increase in net size is countered by the decrease and variance. Using this technique, Cohen-Addad et al. [2022] obtained a coreset of size roughly $\varepsilon^{-2}k\sigma^2$.

To obtain a bound on $\sigma^2$, the framework by Cohen-Addad et al. [2021a] proposed an algorithm that first computes a solution $\mathcal{A}$ with strong structural properties and then samples any point $p$ proportionate $\mathrm{cost}(p, \mathcal{A})$. To simplify the exposition, we assume that $\mathcal{A}$ is the optimum, every cluster has identical cost and every point has identical cost. For this special case, the distribution of Cohen-Addad et al. [2021a] turns out to be equivalent to uniform sampling. For any given solution $\mathcal{S}$, let $k_i$ be the number of clusters of $\mathcal{A}$ whose points are served at cost $2^i$ times their cost in $\mathcal{A}$ for $i > 2^2$, i.e. $2^i = \frac{\mathrm{cost}(p,\mathcal{S})}{\mathrm{cost}(p,\mathcal{A})}$. Cohen-Addad et al. [2021a] showed that their sampling distribution leads to a variance of the order $\sigma^2 \approx \left(\frac{k \cdot k_i 2^i}{(k+k_i \cdot 2^i)^2}\right) \cdot \min(\varepsilon^{-2}, 2^i, k)$, which yields the $\tilde{O}(k\varepsilon^{-2}\min(k,\varepsilon^{-2}))$ bound.

To improve either the variance or the net size has to reduced. Unfortunately, is unlikely that reducing $\sigma^2$ will be possible as the bounds on $\sigma^2$ obtained by Cohen-Addad et al. [2022] are tight up to constant factors. Our main goal is to find a net with an finer error of

$$|\mathrm{cost}(p, \mathcal{S}_\alpha) - \mathrm{cost}(p, \mathcal{S})| \leq 2^{-h} \cdot \sqrt{\mathrm{cost}(p, \mathcal{S}), \mathrm{cost}(p, \mathcal{A})}.$$

---

[1] Coresets of size $\tilde{O}(kd/\varepsilon^2)$ are also known (Cohen-Addad et al. [2021a]). This offers improvements in low dimensions, but generally a dependency on $d$ is considered worse than a dependency on $k$ or $\varepsilon^{-2}$.

[2] Observe that if one points of a cluster $C_j$ of $\mathcal{A}$ costs $2^i$ times more for $i \geq 3$, then all points from $C_j$ do likewise (up to constant factors).

In the case of $\min(2^i, \varepsilon^{-2}, k) = 2^i$, this leads to a reduced variance for the $h$-th summand of the order $2^{-2h} \cdot \frac{(\sqrt{\cost(p,\mathcal{S})\cost(p,\mathcal{A})})^2}{\cost(p,\mathcal{A})^2} \cdot \left(\frac{k \cdot k_i}{(k + k_i \cdot 2^i)^2}\right) \leq 2^{-2h} \cdot \left(\frac{k \cdot k_i \cdot 2^i}{(k + k_i \cdot 2^i)^2}\right)$. To find such a net, we now have two options. First, we can essentially use the previous nets and rescale $2^{-h}$ by a factor $2^{-i/2}$. Unfortunately, this leads to nets of size $\exp(k \cdot 2^{2h}2^i)$, so any gain in reducing the variance is countered by an increase in the net size.

The novelty in our approach now lies in showing that a net of size $\exp(k \cdot k_i \cdot 2^{2h})$ exists. Combining the two net sizes with the improved variance bound results in a coreset of size roughly $\varepsilon^{-2} \log\left(\exp(k \cdot \min(k_i, 2^i) \cdot 2^{2h})\right) \cdot 2^{-2h} \cdot \left(\frac{k \cdot k_i \cdot 2^i}{(k + k_i \cdot 2^i)^2}\right)$, which after some calculation is shown to be of the order $O(k^{1.5}\varepsilon^{-2})$.

For the remainder of this section we will illustrate how these improved nets can be obtained. The key idea already appears in the case of a single center. Suppose that $s$ is a candidate center. Then we can show that there always exists a subspace with orthogonal basis $U$ spanned by a set $T$ of at most $O(\alpha^{-2})$ points of $P$ such that

$$|p^T(I - UU^T)s| \leq \alpha \cdot \|(I - UU^T)p\| \cdot \|(I - UU^T)s\|.$$

We can then write

$$\|p - s\|^2 = \|U(p - s)\|^2 + \|(I - UU^T)p\|^2 + \|(I - UU^T)s\|^2 - 2p^T(I - UU^T)s.$$

The error for the terms $\|Up - s\|^2$ and $\|(I - UU^T)s\|^2$ can be made negligibly small using a net of size $\exp(\alpha^{-2})$. Thus the main loss in error comes from the $p^T(I - UU^T)s$ term. The key insight is that when adding the center $c_i$ of point $p$ in solution $\mathcal{A}$, we have $\|(I - UU^T)p\|^2 \leq \sqrt{\cost(p,\mathcal{A})}$. Thus, when adding all the $k_i$ centers of clusters that cost $2^i$ more in $\mathcal{S}$ than in $\mathcal{A}$ to $T$ the net size becomes $\exp(\alpha^{-2} + k_i)$. By composing these nets for $k$ candidate centers, we then obtain our desired bound.

## 1.2 Related Work

There has been a tremendous amount of work on coresets for Euclidean $k$-means following the work in Bachem et al. [2018a,b], Baker et al. [2020], Bandyapadhyay et al. [2021], Becchetti et al. [2019], Braverman et al. [2022, 2021b], Chen [2009], Cohen-Addad and Li [2019], Cohen-Addad et al. [2021a,b], Feldman and Langberg [2011], Feldman et al. [2020], Feng et al. [2021], Fichtenberger et al. [2013], Har-Peled and Kushal [2007], Har-Peled and Mazumdar [2004], Huang and Vishnoi [2020], Huang et al. [2018, 2019], Langberg and Schulman [2010], Schmidt et al. [2019], Schwiegelshohn and Sheikh-Omar [2022], Sohler and Woodruff [2018]. Almost as prolific is the catalogue of work on dimension reduction for clustering problems in Euclidean spaces, see Boutsidis et al. [2009, 2010, 2015], Charikar and Waingarten [2022a,b], Cohen et al. [2015], Cohen-Addad and Schwiegelshohn [2017], Drineas et al. [2004], Feng et al. [2019], Makarychev et al. [2019]. The arguably most important dimension reduction technique for coresets are terminal embeddings, see Cherapanamjeri and Nelson [2021], Elkin et al. [2017], Mahabadi et al. [2018], Narayanan and Nelson [2019]. Subsequently to our publication, Huang et al. [2022] obtained similar bounds to our result. Specifically, they showed a $\tilde{O}(k^{(2z+2)/z+2}\varepsilon^{-2})$ coreset bound for $(k, z)$ clustering, which recovers our result for $k$-means and $k$-median and extends it to higher powers.

Further work on coresets considering objects other than points as centers Braverman et al. [2021a], Feldman et al. [2010], Huang et al. [2021] or other objectives all together Boutsidis et al. [2013], Huang et al. [2020], Jiang et al. [2021], Karnin and Liberty [2019], Mai et al. [2021], Molina et al. [2018], Munteanu et al. [2018], Phillips and Tai [2020], Tukan et al. [2020]. For further reading, we refer the interested reader to recent surveys Feldman [2020], Munteanu and Schwiegelshohn [2018].

## 2 Preliminaries and Setup

First, we require the following basic notions. For a point $p \in \mathbb{R}^d$, we denote $\|p\|_2 = \sqrt{\sum_{i=1}^d p_i^2}$ to be the Euclidean norm of $p$ and $\|p\|_1 = \sum_{i=1}^d |p_i|$. The distinct number of points in a point set $P$ is denoted by $\|P\|_0$. Note that the true number of points $|P|$ may be larger than $\|P\|_0$ as different points may lie on the same coordinates. Given a solution $\mathcal{S}$ consisting of at most $k$ centers, and any subset $P' \subset P$ we use $\cost(P', \mathcal{S}) := \sum_{p \in P'} \cost(p, \mathcal{S}) = \sum_{p \in P} \min_{s \in \mathcal{S}} w_p \cost(p, s)$, where

$\text{cost}(p, s) = \|p - s\|^2$ for Euclidean $k$-means and $w_p$ is a non-negative weight (in the basic case this simply 1 whereas for the coreset it can be any non-negative number). We also denote by $v^{\mathcal{S}} \in \mathbb{R}^{\|P\|_0}$ the cost vector associated with the point set $P$ and solution $\mathcal{S}$, that is $v_p^{\mathcal{S}} := w_p \text{cost}(p, s)$. Note that $\|v^{\mathcal{S}}\|_1 = \text{cost}(P, \mathcal{S})$. The classic coreset guarantee is to show that for any solution $\mathcal{S}$ the designated coreset $\Omega$ satisfies

$$|\text{cost}(\Omega, \mathcal{S}) - \text{cost}(P, \mathcal{S})| \leq \varepsilon \cdot \text{cost}(P, \mathcal{S}).$$

We will later introduce an equivalent statement that uses cost vectors. It will also be convenient to consider coresets with an additive error $E$ which satisfy

$$|\text{cost}(\Omega, \mathcal{S}) - \text{cost}(P, \mathcal{S})| \leq \varepsilon \cdot \text{cost}(P, \mathcal{S}) + E.$$

Cohen-Addad et al. [2022] showed that any coreset algorithm that works for instances with the following assumptions can be extended to general instances:

**Assumption 1:** $\|P\|_0 \in \text{poly}(k, \varepsilon^{-1})$.

**Assumption 2:** $d \in O(\log(k/\varepsilon) \cdot \varepsilon^{-2})$.

**Assumption 3:** $w_p = 1$, for all $p \in P$. Note that this only applies to the weights of the original points; the coreset points will have different weights.

**Assumption 4:** There exists a solution $\mathcal{A}$ such that

1. $|\mathcal{A}| \in O(k)$.

2. For any two clusters $C_i, C_j$ induced by $\mathcal{A}$, $\text{cost}(C_i, \mathcal{A}) \leq 2 \cdot \text{cost}(C_j, \mathcal{A})$.

3. For any cluster $C_j$ induced by $\mathcal{A}$ and any two points $p, p' \in C_j$, $\text{cost}(p, \mathcal{A}) \leq 2 \cdot \text{cost}(p', \mathcal{A})$

To keep this paper self contained, we will later detail the validity of these assumptions.

The sampling procedure is now very simple. Given that these aforementioned assumptions hold, we sample a points $p \in C_j$ with probability $\mathbb{P}_p := \frac{1}{|C_j|} \cdot \frac{\text{cost}(C_j, \mathcal{A})}{\text{cost}(P, \mathcal{A})}$ and add it to the designated coreset $\Omega$. Furthermore, $p$ receives the weight $w_p = \frac{1}{\mathbb{P}_p}$. Overall, our basic cost estimator for any candidate solution $\mathcal{S}$ is therefore $\text{cost}(\Omega, \mathcal{S}) := \frac{1}{|\Omega|} \sum_{p \in \Omega} \text{cost}(p, \mathcal{S}) \cdot w_p$. It is routine to check that $\mathbb{E}[\text{cost}(\Omega, \mathcal{S})] = \text{cost}(P, \mathcal{S})$. The remainder of this section will be devoted to showing that $\Omega$ satisfies for all $\mathcal{S}$

$$|\text{cost}(\Omega, \mathcal{S}) - \text{cost}(P, \mathcal{S})| \leq \frac{\varepsilon}{\log^2 \varepsilon^{-1}} \cdot (\text{cost}(P, \mathcal{S}) + \text{cost}(P, \mathcal{A})) \tag{1}$$

Using the framework from Cohen-Addad et al. [2021a], this implies an $O(\varepsilon)$ coreset in general.

## 2.1 Justification of the Assumptions

To obtain the first assumption, we compute any coreset of size $\text{poly}(k, \varepsilon^{-1})$ in preprocessing. Constructions of these coresets are abundant in literature and any one would serve our needs.

To obtain the second assumption, we apply a terminal embedding on the coreset. A terminal embedding guarantees that for any point $p \in P$ and any point $q \in \mathbb{R}^d$, where $d$ is the dimension of the points of $P$, we have a mapping $f$ s.t.

$$\|p - q\|^2 = (1 \pm \varepsilon)\|f(p) - f(q)\|^2.$$

Narayanan and Nelson [2019] showed that for any $n$-point set a terminal embedding of target dimension $\tilde{O}(\varepsilon^{-2} \log n)$ exists, which, combined with the first assumption, yields the desired target dimension.

To obtain the third assumption, we merely have to ensure that the weights of the coreset points are integers. A number of constructions satisfy this but a simple way of always enforcing this is to scale and round the weights (see Corollary 2 of Cohen-Addad et al. [2021a]).

The fourth assumption follows from the preprocessing of Cohen-Addad et al. [2021a], see Sections 3.3 and 4.1 of that reference. Similarly, the same preprocessing, given that $\mathcal{A}$ is an $O(1)$-approximation, also shows that Eq 1 implies that the overall construction will be a coreset (subject to rescaling $\varepsilon$ by constant factors), see Section 4.2 of the aforementioned reference. We must point out that a point set

cannot always be decomposed into only sets that satisfy the aforementioned assumption. Nevertheless Cohen-Addad et al. [2022] showed that every other case require only $\tilde{O}(k/\varepsilon^2)$ many sampled points (compared Lemmas 15 and 17 of that reference.)

Finally, we remark that these steps and assumptions immediately also apply to the $k$-median problem.

## 3   Analysis

In this section we prove the following theorem.

**Theorem 1.** *For any set of points in $d$ dimensional Euclidean space, there exists a coreset for $k$-means clustering of size $\tilde{O}(k^{1.5}\varepsilon^{-2})$.*

For the $k$-median case, we also obtain an improved bound as follows. Details will mainly consider the result for $k$-means. The analysis for $k$-median is given in the supplementary material.

**Theorem 2.** *For any set of points in $d$ dimensional Euclidean space, there exists a coreset for $k$-means clustering of size $\tilde{O}(k^{4/3}\varepsilon^{-2})$.*

We first describe the random process used to show concentration of the estimator.

### 3.1   Setting up the Chaining Analysis

First, we observe that Eq.1 is equivalent to showing

$$\sup_{\mathcal{S}} \frac{|\text{cost}(\Omega, \mathcal{S}) - \|v\|_1|}{(\text{cost}(P, \mathcal{S}) + \text{cost}(P, \mathcal{A}))} \leq \frac{\varepsilon}{\log^2 \varepsilon^{-1}}.$$

Our goal is to show that

$$\mathbb{E}_\Omega \left[ \sup_{\mathcal{S}} \frac{|\text{cost}(\Omega, \mathcal{S}) - \|v\|_1|}{(\text{cost}(P, \mathcal{S}) + \text{cost}(P, \mathcal{A}))} \right] \leq \frac{\varepsilon}{\log^2 \varepsilon^{-1}},$$

where $\mathbb{E}_\Omega$ is meant to denote the expectation over the randomness of $\Omega$. This implies that the desired guarantee holds with constant probability.

We now apply a standard symmetrization argument.

**Lemma 1** (Appendix B.3 of Rudra and Wootters [2014]). *Let $g_p$ be independent standard Gaussian random variables. Then.*

$$\mathbb{E}_\Omega \sup_{\mathcal{S}} \left[ \left| \frac{\frac{1}{|\Omega|} \sum_{p \in \Omega} cost(p, \mathcal{S}) \cdot w_p - \|v\|_1}{(cost(P, \mathcal{S}) + cost(P, \mathcal{A}))} \right| \right] \leq \sqrt{2\pi} \mathbb{E}_\Omega \mathbb{E}_g \sup_{\mathcal{S}} \left[ \left| \frac{\frac{1}{|\Omega|} \sum_{p \in \Omega} cost(p, \mathcal{S}) \cdot w_p \cdot g_p}{(cost(P, \mathcal{S}) + cost(P, \mathcal{A}))} \right| \right].$$

It is therefore sufficient to show

$$\mathbb{E}_\Omega \mathbb{E}_g \sup_{\mathcal{S}} \left[ \left| \frac{\frac{1}{|\Omega|} \sum_{p \in \Omega} \text{cost}(p, \mathcal{S}) \cdot w_p \cdot g_p}{(\text{cost}(P, \mathcal{S}) + \text{cost}(P, \mathcal{A}))} \right| \right] \leq \frac{\varepsilon}{\sqrt{2\pi} \log^2 \varepsilon^{-1}}. \tag{2}$$

We partition the clusters of any solution $\mathcal{S}$ by type. We consider a cluster $C_j$ of type $T_i$ if

$$2^i \min_{c \in \mathcal{A}} \|p - c\|^2 \leq \min_{p \in C_j} \min_{s \in \mathcal{S}} \|p - s\|^2 \leq 2^{i+1} \min_{c \in \mathcal{A}} \|p - c\|^2.$$

The number of clusters $C_j \in T_i$ are denoted by $k_i$. If $C_j$ is of type $i \leq 3$, we say $C_j$ is of type $T_{small}$ and if $C_j$ is of type $i \geq \log \gamma \varepsilon^{-2}$, for a sufficiently large absolute constant $\gamma$, we say that $C_j$ is of type $T_{large}$. Then, we show

$$\mathbb{E}_\Omega \mathbb{E}_g \left[ \sup_{\mathcal{S}} \left| \frac{\frac{1}{|\Omega|} \sum_{C_j \in T_{small}} \sum_{p \in C_j \cap \Omega} \text{cost}(p, \mathcal{S}) w_p \cdot g_p}{(\text{cost}(P, \mathcal{S}) + \text{cost}(P, \mathcal{A}))} \right| \right] \leq \frac{\varepsilon}{\sqrt{2\pi} \log^3 \varepsilon^{-1}} \tag{3}$$

$$\mathbb{E}_\Omega \mathbb{E}_g \left[ \sup_{\mathcal{S}} \left| \frac{\frac{1}{|\Omega|} \sum_{C_j \in T_i} \sum_{p \in C_j \cap \Omega} \text{cost}(p, \mathcal{S}) w_p \cdot g_p}{(\text{cost}(P, \mathcal{S}) + \text{cost}(P, \mathcal{A}))} \right| \right] \leq \frac{\varepsilon}{\sqrt{2\pi} \log^3 \varepsilon^{-1}} \tag{4}$$

$$\mathbb{E}_\Omega \mathbb{E}_g \left[ \sup_{\mathcal{S}} \left| \frac{\frac{1}{|\Omega|} \sum_{C_j \in T_{large}} \sum_{p \in C_j \cap \Omega} \text{cost}(p, \mathcal{S}) w_p \cdot g_p}{(\text{cost}(P, \mathcal{S}) + \text{cost}(P, \mathcal{A}))} \right| \right] \leq \frac{\varepsilon}{\sqrt{2\pi} \log^3 \varepsilon^{-1}} \tag{5}$$

Note that if Equation 4 holds for $i \in \{3, \ldots, \log 1/\varepsilon\}$, this also implies Equation 2, as the error from each type can only sum up in the worst case and there are at most $O(\log \varepsilon^{-1})$ many types.

The small and large types are comparatively simple to handle.

**Lemma 2** (Lemmas 15 and 16 of Cohen-Addad et al. [2022])**.** *Let $|\Omega| \geq \kappa \frac{k}{\varepsilon^2} polylog(k/\varepsilon)$ for some absolute constant $\kappa$. Then Equations 3 and 5 hold.*

Our main objective will be to prove the following lemma.

**Lemma 3.** *Let $|\Omega| \geq \kappa \cdot \frac{k^{1.5}}{\varepsilon^2} polylog(k/\varepsilon)$ for some absolute constants $\kappa$. Then Equation 4 holds.*

Combining Lemma 2 and Lemma 3 then implies Theorem 2.

### 3.2 Sketch of the Proof of Lemma 3

The proof of Lemma 3 mainly consists of defining a nested sequence of nets over cost vectors over which we apply a union bound. Roughly speaking, for any cost vector $v^{\mathcal{S}}$, we aim to find an approximating cost vector $v'$ such that

$$\text{cost}(p, \mathcal{S}) = v_p^{\mathcal{S}} - v_p' \leq \varepsilon \cdot \sqrt{\text{cost}(p, \mathcal{S}) \cdot \text{cost}(p, \mathcal{A})}.$$

This analysis differs from the terminal-embedding-based nets one used in Cohen-Addad et al. [2022], which aimed for an error of the order $\varepsilon \cdot \text{cost}(p, \mathcal{S})$.

Suppose we have, for every $\varepsilon$, a suitable collection of approximating cost vectors $\mathbb{N}_{\log 1/\varepsilon}$ with this guarantee for any candidate $\mathcal{S}^3$. Let $v^{\mathcal{S},\varepsilon}$ be the cost vector approximating $v^{\mathcal{S}}$ in the net $\mathbb{N}_{\log 1/\varepsilon}$. Then we can write

$$v_p^{\mathcal{S}} = \sum_{h=0}^{\infty} v_p^{\mathcal{S},2^{-(h+1)}} - v_p^{\mathcal{S},2^{-h}},$$

with $v_p^{\mathcal{S},1} = 0$. Our goal is to now bound

$$\mathbb{E}_\Omega \mathbb{E}_g \left[ \sup_{\mathcal{S}} \left| \frac{\sum_{C_j \in T_i} \sum_{p \in C_j \cap \Omega} \text{cost}(p, \mathcal{S}) w_p \cdot g_p}{|\Omega| \cdot (\text{cost}(P, \mathcal{S}) + \text{cost}(P, \mathcal{A}))} \right| \right]$$

$$= \mathbb{E}_\Omega \mathbb{E}_g \left[ \sup_{v^{\mathcal{S}}} \left| \frac{\sum_{h=0}^{\infty} \sum_{C_j \in T_i} \sum_{p \in C_j \cap \Omega} (v_p^{\mathcal{S},2^{-(h+1)}} - v_p^{\mathcal{S},2^{-h}}) w_p \cdot g_p}{|\Omega| \cdot (\text{cost}(P, \mathcal{S}) + \text{cost}(P, \mathcal{A}))} \right| \right]$$

$$\leq \sum_{h=0}^{\infty} \mathbb{E}_\Omega \mathbb{E}_g \left[ \sup_{v^{\mathcal{S},h+1} - v^{\mathcal{S},h} \in \mathbb{N}_{2^{-(h+1)}} \times \mathbb{N}_{2^{-h}}} \left| \frac{\sum_{C_j \in T_i} \sum_{p \in C_j \cap \Omega} (v_p^{\mathcal{S},2^{-(h+1)}} - v_p^{\mathcal{S},2^{-h}}) w_p \cdot g_p}{|\Omega| \cdot (\text{cost}(P, \mathcal{S}) + \text{cost}(P, \mathcal{A}))} \right| \right]$$

$$= \mathbb{E}_\Omega \mathbb{E}_g \left[ \sup_{v^{\mathcal{S},1} \in \mathbb{N}_{2^{-1}}} \left| \frac{\sum_{C_j \in T_i} \sum_{p \in C_j \cap \Omega} v_p^{\mathcal{S},2^{-1}} w_p \cdot g_p}{|\Omega| \cdot (\text{cost}(P, \mathcal{S}) + \text{cost}(P, \mathcal{A}))} \right| \right] \tag{6}$$

$$+ \sum_{h=1}^{\log \varepsilon^{-2}} \mathbb{E}_\Omega \mathbb{E}_g \left[ \sup_{v^{\mathcal{S},h+1} - v^{\mathcal{S},h} \in \mathbb{N}_{h+1} \times \mathbb{N}_h} \left| \frac{\sum_{C_j \in T_i} \sum_{p \in C_j \cap \Omega} (v_p^{\mathcal{S},2^{-(h+1)}} - v_p^{\mathcal{S},2^{-h}}) w_p \cdot g_p}{|\Omega| \cdot (\text{cost}(P, \mathcal{S}) + \text{cost}(P, \mathcal{A}))} \right| \right] \tag{7}$$

$$+ \sum_{\log \varepsilon^{-2}}^{\infty} \mathbb{E}_\Omega \mathbb{E}_g \left[ \sup_{v^{\mathcal{S},h+1} - v^{\mathcal{S},h} \in \mathbb{N}_{h+1} \times \mathbb{N}_h} \left| \frac{\sum_{C_j \in T_i} \sum_{p \in C_j \cap \Omega} (v_p^{\mathcal{S},2^{-(h+1)}} - v_p^{\mathcal{S},2^{-h}}) w_p \cdot g_p}{|\Omega| \cdot (\text{cost}(P, \mathcal{S}) + \text{cost}(P, \mathcal{A}))} \right| \right] \tag{8}$$

We will bound Equations 6 and 8 directly. For the $O(\log \varepsilon^{-1})$ equations in term 7, we prove a bound on each. Thus, we aim for a bound of the order $O(\frac{\varepsilon}{\log^3 \varepsilon^{-1}})$; the overall bound then follows by summing up the errors and rescaling by constant factors. Technically, bounding each of the terms in Equations 6, 7 and 8 requires somewhat different arguments. For the sake of illustrating the key new ideas we focus on Eq. 7. A full proof is contained in the supplementary material.

The next section presents the nets for the cost vectors. The subsequent section bounds the variance. The final section combines these results and completes the proof of Lemma 3.

---

[3]The reason for indexing the net by $\mathbb{N}_{\log 1/\varepsilon}$ and not by $\mathbb{N}_\varepsilon$ is to conveniently sum over $\sum_{i=1}^{\infty} \log |N_i|$, rather than $\sum_{i=1}^{\infty} \log |N_{2^{-i}}|$.

**Cost Vector Nets**

**Definition 1.** *Let $I$ be a metric space, $P$ a set of points, $k$ a positive integer, and let $\alpha > 0$ be a precision parameters and let $\mathcal{A}$ be some solution with at most $k'$ centers. Let $\mathbb{C} \subset I^k$ be a (potentially infinite) set of candidate $k$-clusterings. We say that a set of cost vectors $\mathbb{N} \subset \mathbb{R}^{|P|}$ is an $(\alpha, k)$-means clustering net if for every $\mathcal{S} \in \mathbb{C}$ there exists a vector $v' \in \mathbb{N}$ such that the following condition holds. For all $p \in P$,*

$$|v_p^{\mathcal{S}} - v_p'| \le \alpha \cdot (\sqrt{cost(p, \mathcal{S})cost(p, \mathcal{A})} + cost(p, \mathcal{A})).$$

These nets have a substantially smaller error than those proposed in Cohen-Addad et al. [2022], which had an error of the order $\alpha \cdot (\text{cost}(p, \mathcal{S}) + \text{cost}(p, \mathcal{A}))$.

Given a set of points $X$ in Euclidean space, an $\varepsilon$-net is a subset $S \subset X$ such that for every $p \in X$ there exists a $q$ in $S$ with $\|p - q\| \le \varepsilon$. Throughout this section, we will frequently use the fact that in $d$ dimensions, there exists an $\varepsilon$-net of cardinality $(1 + 2/\varepsilon)^d$ (see for example Pisier [1999]). Our main goal in this section is to prove the following lemma.

**Lemma 4.** *Let $P$ be a set of points in $d$ dimensional Euclidean space, $k$ a positive integer, $\mathcal{A}$ be a candidate solution with $k_i$ clusters and $\gamma$ an absolute constant. Define $\mathbb{C}$ to be the set of possible candidate centers such that a subset $A_i$ the clusters induced by $\mathcal{A}$ are of type $i$, with $3 \le i \le \log 1/\varepsilon^2$. For all $\alpha \le 1/2$, there exists an $(\alpha, k)$-means clustering net $\mathbb{N}$ of $\mathbb{C}$ with*

$$|\mathbb{N}| \le \exp\left( \gamma \cdot k \cdot \log \|P\|_0 \cdot \min(k_i + \alpha^{-2}, \alpha^{-2} \cdot 2^i) \cdot i \log \frac{1}{\alpha} \right).$$

*Proof (Sketch).* We first show that given a set of vectors $P$ and any vector $s$, there always exists a small subset $U$ of $P$ such that all inner products between $p \in P$ and $s$ are preserved by the span of $U$.

**Lemma 5.** *Let $P = \{p_1, ..., p_n\} \subseteq \mathbb{R}^d$ and let $s \in \mathbb{R}^d$. Then there exists $U \subseteq P$, with $|U| = O(\varepsilon^{-2})$ and orthogonal basis $\Pi_U$, such that*

$$\forall p \in P, |p^T (I - \Pi_U \Pi_U^T)s| \le \varepsilon \|(I - \Pi_U \Pi_U^T)p\| \cdot \min_{p \in P} \|p - s\| \tag{9}$$

*Proof.* Start with $U_0 = \text{argmin}_{p \in P} \|p - s\|$, and proceed in rounds. Note that $\|(I - \Pi_{U_0} \Pi_{U_0}^T)s\| \le \|p - s\|$ for all $p \in P$.

In each round $i$, denote the current set of vectors $U_i$ with orthogonal basis $\Pi_{U_i}$. We add a vector $p_i$ if the following equation holds

$$|p^T (I - \Pi_{U_i} \Pi_{U_i}^T)s| \ge \varepsilon \|(I - \Pi_{U_i} \Pi_{U_i}^T)p\| \cdot \|(I - \Pi_{U_0} \Pi_{U_0}^T)s\|.$$

We observe that if this equation holds for all $p \in P$, then Equation 9 must also hold. Note that $(I - \Pi_{U_i} \Pi_{U_i}^T)p$ is orthogonal to the span of $U_i$ of all previously added vectors. Thus, due to the Pythagorean theorem, we have

$$\sum_i^t \left( \frac{(p^T (I - \Pi_{U_{i-1}} \Pi_{U_{i-1}}^T)s)}{\|(I - \Pi_{U_{i-1}} \Pi_{U_{i-1}}^T)p\| \cdot \|(I - U_0 U_0^T)s\|} \right)^2 \ge t \cdot \varepsilon^2.$$

Therefore, after $t = \varepsilon^{-2}$ many rounds $(I - \Pi_U \Pi_U^T)s = 0$, which implies that after at most $\varepsilon^{-2}$ rounds Eq. 9 has to hold. $\square$

With this lemma, we can prove our net bound. Our objective is to generate a small set of cost vectors that satisfy the desired guarantee. We first define the cost vectors. For each subset $U$ of size $O(\min(\alpha^{-2}2^i, \alpha^{-2} + k_i)$, we consider the the subspace $\Pi_U$ spanned by $U$. In this subspace we consider $(\alpha/2^i) \cdot \sqrt{\text{cost}(p, \mathcal{A})}$-nets of every ball centered around $\Pi_U p$ with radius $60 \cdot 2^i/2 \cdot \sqrt{\text{cost}(p, \mathcal{A})}$ for all $p \in P$. Such a net has size $\exp(\gamma \cdot \textbf{rank}(U)i \log \alpha)$, for some constant $\gamma$ and there exist at most $\|P\|_0 \cdot \exp(\gamma \cdot |U|i \log \alpha)$ many such nets. Furthermore, there are at most $\binom{\|P\|_0}{|U|} \le \|P\|_0^{|U|}$ such subsets.

Now, for every point $p$, define an exponential sequence $\alpha^2(1 + \alpha/2^i)^j$ for $j \in \{0, \dots \log 10 \cdot 2^i\}$. There exist at most $\|P\|_0$ such sequences and every such sequence consists of at most $O(\alpha^{-1} \cdot 2^i \cdot i)$

many values. We combine every net point in ever ball of every subspace with all values in the exponential sequence to obtain the evaluation for a single candidate center. The overall number of candidate centers is therefore of the order $\|P\|_0^{|U|} \cdot \exp(\gamma \cdot |U| i \log \alpha)$, for a sufficiently large $\gamma$. The overall number of candidate cost vectors is now the number of $k$ subsets of candidate centers, i.e. $\|P\|_0^{k \cdot |U|} \cdot \exp(\gamma \cdot k \cdot |U| i \log \alpha)$. Combined with the bounds on $U$, this yields the desired size. What remains to be shown is that the thus constructed cost vectors are a $(\alpha, k)$-clustering net.

Here, we use that for any center $s$ in some candidate solution $\mathcal{S}$

$$\|p - s\|^2 = \|\Pi_U(p - s)\|^2 + \|(I - \Pi_U \Pi_U^T)p\|^2 + \|(I - \Pi_U \Pi_U^T)s\|^2 - 2p^T(I - \Pi_U \Pi_U^T)s.$$

The nets for the span of $\Pi_U$ are so fine that the distance $\|\Pi_U s - s'\|^2$ is essentially negligible, where $s'$ is the point in the span of $\Pi_U$ closest to $\Pi_U s$ and the same holds for the exponential sequence approximating the term $\|(I - \Pi_U \Pi_U^T)s\|^2$. Thus, the error is dominated by $2p^T(I - \Pi_U \Pi_U^T)s$. Now, we can assume that the input point closest to $s$ is included in $U$. Then $\min_{p \in P} \|p - s\| \leq \sqrt{\mathrm{cost}(p, \mathcal{S})} \leq O(1) \cdot 2^{i/2} \sqrt{\mathrm{cost}(p, \mathcal{A})}$ and $\|(I - \Pi_U \Pi_U^T)p\| \leq \sqrt{\mathrm{cost}(p, \mathcal{S})} \leq O(1) \cdot 2^{i/2} \sqrt{\mathrm{cost}(p, \mathcal{A})}$. If $\alpha^{-2} \cdot 2^i < k_i + \alpha^{-2}$, we have

$$|p^T(I - \Pi_U \Pi_U^T)s| \leq \alpha \cdot 2^{-i/2} \cdot \|(I - \Pi_U \Pi_U^T)p\| \cdot \min_{p \in P} \|p - s\| \leq O(1) \cdot \alpha \cdot \sqrt{\mathrm{cost}(p, \mathcal{S})\mathrm{cost}(p, \mathcal{A})}$$

otherwise we have $\|(I - \Pi_U \Pi_U^T)p\| \leq \sqrt{\mathrm{cost}(p, \mathcal{A})}$ which implies

$$|p^T(I - \Pi_U \Pi_U^T)s| \leq \alpha \cdot \|(I - \Pi_U \Pi_U^T)p\| \cdot \min_{p \in P} \|p - s\| \leq \alpha \cdot \sqrt{\mathrm{cost}(p, \mathcal{S})\mathrm{cost}(p, \mathcal{A})}.$$

Rescaling $\alpha$ by constant factors yields the claim. $\qquad\square$

**Corollary 3.** *Let $P$ be a set of points in $d$ dimensional Euclidean space, $k$ a positive integer, $\mathcal{A}$ be a candidate solution with $O(k)$ clusters and $\gamma$ an absolute constant. Then there exists an $(\alpha, k)$ clustering net of size Define $\mathbb{C}$ to be the set of possible candidate centers. such that a subset $A_i$ the clusters induced by $\mathcal{A}$ are of type $i$, with $3 \leq i \leq \log 1/\varepsilon^2$. For all $\alpha \leq 1/2$, there exists an $(\alpha, k)$-means clustering net $\mathbb{N}$ of $\mathbb{C}$ with*

$$|\mathbb{N}| \leq \exp\left(\gamma \cdot k \cdot \log \|P\|_0 \cdot \min(k_i + \alpha^{-2}, \alpha^{-2} \cdot 2^i) \cdot i \log \frac{1}{\alpha})\right).$$

**Bounding the Variance**

We now use the cost vectors to obtain an improved variance for the estimator $\frac{\sum_{C_j \in T_i} \sum_{p \in C_j \cap \Omega}(v_p^{\mathcal{S}, 2^{-(h+1)}} - v_p^{\mathcal{S}, 2^{-h}})w_p}{(\mathrm{cost}(P, \mathcal{S}) + \mathrm{cost}(P, \mathcal{A}))} g_p$. The bounds on variance for any random variable $\sum a_p g_p$ with standard Gaussians $g_p$ is Gaussian distributed with mean $0$ and variance $\sum a_p^2$.

Before we do this, we require an additional notion. Let $\mathcal{E}$ denote the event that $\frac{1}{|\Omega|} \sum_{p \in C_j \cap \Omega} w_p = (1 \pm \varepsilon) \cdot |C_j|$. The following lemma bounds the probability of $\mathcal{E}$ occurring.

**Lemma 6.** *[Compare Lemma 19 of Cohen-Addad et al. [2022]] If Assumption 4 holds, then event $\mathcal{E}$ holds with probability $1 - k^{-2}$ if $|\Omega| > \kappa \cdot k \varepsilon^{-2} \log k$ for a sufficiently high absolute constant $\kappa$.*

**Lemma 7.** *Given Assumption 4, the variance of $\frac{\sum_{C_j \in T_i} \sum_{p \in C_j \cap \Omega}(v_p^{\mathcal{S}, 2^{-(h+1)}} - v_p^{\mathcal{S}, 2^{-h}})w_p \cdot g_p}{|\Omega| \cdot (\mathrm{cost}(P, \mathcal{S}) + \mathrm{cost}(P, \mathcal{A}))}$ is at most*

$$\gamma \cdot \frac{2^{-2h}}{|\Omega|} \cdot \frac{k \cdot k_i \cdot 2^i}{(k + k_i \cdot 2^i)^2} \qquad \text{conditioned on event } \mathcal{E}$$

$$\gamma \cdot \frac{2^{-2h} \cdot k}{|\Omega|} \cdot \frac{k \cdot k_i \cdot 2^i}{(k + k_i \cdot 2^i)^2} \qquad \text{conditioned on event } \overline{\mathcal{E}}$$

*for an absolute constant $\gamma$.*

*Proof.* We first observe that since the $g_p$ are standard normal Gaussians, the entire estimator is Gaussian distributed with variance

$$\sum_{C_j \in T_i} \sum_{p \in C_j \cap \Omega} \frac{1}{|\Omega|^2} \left(\frac{(v_p^{\mathcal{S}, 2^{-(h+1)}} - v_p^{\mathcal{S}, 2^{-h}})w_p}{(\mathrm{cost}(P, \mathcal{S}) + \mathrm{cost}(P, \mathcal{A}))}\right)^2.$$

We have $(v_p^{\mathcal{S},2^{-(h+1)}} - v_p^{\mathcal{S},2^{-h}}) = (v_p^{\mathcal{S},2^{-(h+1)}} - \mathrm{cost}(p,\mathcal{S}) + \mathrm{cost}(p,\mathcal{S}) - v_p^{\mathcal{S},2^{-h}}) \leq 2 \cdot 2^{-h} \cdot$
$\sqrt{\mathrm{cost}(p,\mathcal{A})\mathrm{cost}(p,\mathcal{S})}$ due to Lemma 4. Furthermore, by definition $w_p = \frac{\mathrm{cost}(P,\mathcal{A})|C_j|}{\mathrm{cost}(C_j,\mathcal{A})}$. Finally, by definition of type $i$, we have $\mathrm{cost}(p,\mathcal{S}) \cdot |C_j| = O(1) \cdot \mathrm{cost}(C_j,\mathcal{S})$ and by Assumption 4 we have $\mathrm{cost}(p,\mathcal{A}) \cdot |C_j| = O(1) \cdot \mathrm{cost}(C_j,\mathcal{A})$ for all $p \in C_j$.

$$\sum_{C_j \in T_i} \sum_{p \in C_j \cap \Omega} \frac{1}{|\Omega|^2} \left( \frac{(v_p^{\mathcal{S},2^{-(h+1)}} - v_p^{\mathcal{S},2^{-h}})w_p}{(\mathrm{cost}(P,\mathcal{S}) + \mathrm{cost}(P,\mathcal{A}))} \right)^2$$

$$\leq O(1) \cdot \sum_{C_j \in T_i} \sum_{p \in C_j \cap \Omega} \frac{1}{|\Omega|^2} \left( \frac{2^{-h} \cdot \sqrt{\mathrm{cost}(p,\mathcal{A})\mathrm{cost}(p,\mathcal{S})} \cdot \mathrm{cost}(P,\mathcal{A})|C_j|}{\mathrm{cost}(C_j,\mathcal{A}) \cdot (\mathrm{cost}(P,\mathcal{S}) + \mathrm{cost}(P,\mathcal{A}))} \right)^2$$

$$\leq O(1) \cdot \sum_{C_j \in T_i} \sum_{p \in C_j \cap \Omega} \frac{1}{|\Omega|^2} \left( \frac{2^{-2h} \cdot \mathrm{cost}(C_j,\mathcal{A}) \cdot \mathrm{cost}(C_j,\mathcal{S})\mathrm{cost}(P,\mathcal{A})^2}{\mathrm{cost}(C_j,\mathcal{A})^2 \cdot (\mathrm{cost}(P,\mathcal{S}) + \mathrm{cost}(P,\mathcal{A}))^2} \right)$$

Now, let $k_i$ be the number of clusters of type $i$. Then due to Assumption 4 $\mathrm{cost}(C_j,\mathcal{S}) \cdot k_i \leq O(1)\mathrm{cost}(P,\mathcal{S})$, for all $C_j$ of type $i$. Finally, note that $\frac{\mathrm{cost}(P,\mathcal{A})}{\mathrm{cost}(C_j,\mathcal{A})} \leq O(1) \cdot k$, also due to Assumption 4. Combining this, we then have

$$\sum_{C_j \in T_i} \sum_{p \in C_j \cap \Omega} \left( \frac{(v_p^{\mathcal{S},2^{-(h+1)}} - v_p^{\mathcal{S},2^{-h}})w_p}{(\mathrm{cost}(P,\mathcal{S}) + \mathrm{cost}(P,\mathcal{A}))} \right)^2$$

$$\leq O(1) \cdot \sum_{C_j \in T_i} \sum_{p \in C_j \cap \Omega} \left( \frac{2^{-2h} \cdot \mathrm{cost}(P,\mathcal{S})\mathrm{cost}(P,\mathcal{A}) \cdot k}{k_i \cdot |\Omega|^2 \cdot (\mathrm{cost}(P,\mathcal{S}) + \mathrm{cost}(P,\mathcal{A}))^2} \right)$$

$$\leq O(1) \cdot \sum_{C_j \in T_i} \sum_{p \in C_j \cap \Omega} \left( \frac{2^{-2h} \cdot k}{k_i \cdot |\Omega|^2} \right) \cdot \frac{k \cdot k_i \cdot 2^i}{(k + k_i \cdot 2^i)^2}$$

Assuming event $\mathcal{E}$, this may now be bounded by $O(1) \cdot \frac{2^{-2h}}{|\Omega|} \cdot \frac{k \cdot k_i \cdot 2^i}{(k+k_i \cdot 2^i)^2}$. If event $\mathcal{E}$ does not hold, we may bound the term by $\frac{2^{-2h} \cdot k}{k_i \cdot |\Omega|} \cdot \frac{k \cdot k_i \cdot 2^i}{(k+k_i \cdot 2^i)^2} \leq \frac{2^{-2h} \cdot k}{|\Omega|} \cdot \frac{k \cdot k_i \cdot 2^i}{(k+k_i \cdot 2^i)^2}$. $\qquad\square$

**Completing the Proof (for Eq. 7)**

Throughout this section, we use the bound on the expected maximum of independent Gaussians.

**Lemma 8** (Lemma 2.3 of Massart [2007]). *Let $g_i \sim \mathcal{N}(0, \sigma_i^2)$, $i \in [n]$ be Gaussian random variables and suppose $\sigma_i \leq \sigma$ for all $i$. Then $\mathbb{E}[\max_{i\in[n]}|g_i|] \leq 2\sigma \cdot \sqrt{2\ln n}$.*

We construct nets for all cost vectors constrained to the entries associated with points of clusters of type $i$ in some solution, i.e. we only consider points $p \in C_j \in T_i$ and the entries $v_p^{\mathcal{S}}$.

For every fixed set of clusters of size $k_i$ of type $T_i$, the number of cost vectors in $\mathbb{N}_{h+1} \times \mathbb{N}_h$ is at most $\exp\left(\gamma \cdot k \cdot \log\|P\|_0 \cdot \min(k_i + 2^{2h}, 2^{2h} \cdot 2^i) \cdot i \cdot h)\right)$ for some absolute constant $\gamma$ due to Lemma 4. We split the vectors in the nets by $k_i \leq \sqrt{k}$ or by $k_i \geq \sqrt{k}$. In both cases, we have $\min(k_i, 2^i) \frac{k \cdot k_i \cdot 2^i}{(k+k_i \cdot 2^i)^2} \leq \sqrt{k}$. With the bound on the variance (Lemma 7 and conditioned on event $\mathcal{E}$), we then have

$$\sum_{h=1}^{\log \varepsilon^{-2}} \mathbb{E}_\Omega \mathbb{E}_g \left[ \sup_{v^{\mathcal{S},h+1} - v^{\mathcal{S},h} \in \mathbb{N}_{h+1} \times \mathbb{N}_h} \left| \frac{\sum_{C_j \in T_i} \sum_{p \in C_j \cap \Omega} (v_p^{\mathcal{S},2^{-(h+1)}} - v_p^{\mathcal{S},2^{-h}})w_p \cdot g_p}{(\mathrm{cost}(P,\mathcal{S}) + \mathrm{cost}(P,\mathcal{A}))} \right| \middle| \mathcal{E} \right]$$

$$
\begin{aligned}
\leq \quad & \sum_{h=1}^{\log \varepsilon^{-2}} \mathbb{E}_\Omega \mathbb{E}_g \left[ \sup_{v^{\mathcal{S},h+1}-v^{\mathcal{S},h} \in \mathbb{N}_{h+1} \times \mathbb{N}_h, k_i \leq \sqrt{k}} \left| \frac{\sum_{C_j \in T_i} \sum_{p \in C_j \cap \Omega}(v_p^{\mathcal{S},2^{-(h+1)}} - v_p^{\mathcal{S},2^{-h}})w_p \cdot g_p}{(\mathrm{cost}(P,\mathcal{S}) + \mathrm{cost}(P,\mathcal{A}))} \right| \, \Big| \, \mathcal{E} \right] \\
+ \quad & \sum_{h=1}^{\log \varepsilon^{-2}} \mathbb{E}_\Omega \mathbb{E}_g \left[ \sup_{v^{\mathcal{S},h+1}-v^{\mathcal{S},h} \in \mathbb{N}_{h+1} \times \mathbb{N}_h, k_i \geq \sqrt{k}} \left| \frac{\sum_{C_j \in T_i} \sum_{p \in C_j \cap \Omega}(v_p^{\mathcal{S},2^{-(h+1)}} - v_p^{\mathcal{S},2^{-h}})w_p \cdot g_p}{(\mathrm{cost}(P,\mathcal{S}) + \mathrm{cost}(P,\mathcal{A}))} \right| \, \Big| \, \mathcal{E} \right] \\
\leq \quad & 2 \sum_{h=1}^{\log \varepsilon^{-2}} \sqrt{\gamma \cdot k \cdot \log \|P\|_0 \cdot 2^{2h} \sqrt{k} \cdot i \cdot h \cdot \frac{2^{-2h}}{|\Omega|}} \\
\leq \quad & 4 \sqrt{\gamma \cdot k\sqrt{k} \cdot \log \|P\|_0 \cdot i \cdot \log^3 \varepsilon^{-1} \cdot \frac{1}{|\Omega|}}
\end{aligned}
\tag{10}
$$

If $\mathcal{E}$ does not hold, we use the somewhat coarse bound of $\min(k_i, 2^i)\frac{k \cdot k_t \cdot 2^i}{(k+k_t \cdot 2^i)^2} \leq k$ to obtain

$$
\begin{aligned}
& \sum_{h=1}^{\log \varepsilon^{-2}} \mathbb{E}_\Omega \mathbb{E}_g \left[ \sup_{v^{\mathcal{S},h+1}-v^{\mathcal{S},h} \in \mathbb{N}_{h+1} \times \mathbb{N}_h} \left| \frac{\sum_{C_j \in T_i} \sum_{p \in C_j \cap \Omega}(v_p^{\mathcal{S},2^{-(h+1)}} - v_p^{\mathcal{S},2^{-h}})w_p \cdot g_p}{(\mathrm{cost}(P,\mathcal{S}) + \mathrm{cost}(P,\mathcal{A}))} \right| \, \Big| \, \overline{\mathcal{E}} \right] \\
\leq \quad & 2 \sqrt{\gamma \cdot k^3 \cdot \log \|P\|_0 \cdot \min(k_i, 2^i) \cdot i \cdot \log^3 \varepsilon^{-1} \cdot \frac{1}{|\Omega|}}.
\end{aligned}
\tag{11}
$$

We have $\mathbb{P}[\overline{\mathcal{E}}] \leq 1/k^2$ due to Lemma 6. Since $\|P\|_0 \leq \mathrm{poly}(k, \varepsilon^{-1})$, $2^i \leq O(1) \cdot \varepsilon^{-2}$ and by our choice of $|\Omega|$, we can combine Equations 10 and 11 with the law of total expectation to obtain

$$
\begin{aligned}
& \sum_{h=1}^{\log \varepsilon^{-2}} \mathbb{E}_\Omega \mathbb{E}_g \left[ \sup_{v^{\mathcal{S},h+1}-v^{\mathcal{S},h} \in \mathbb{N}_{h+1} \times \mathbb{N}_h} \left| \frac{\sum_{C_j \in T_i} \sum_{p \in C_j \cap \Omega}(v_p^{\mathcal{S},2^{-(h+1)}} - v_p^{\mathcal{S},2^{-h}})w_p \cdot g_p}{(\mathrm{cost}(P,\mathcal{S}) + \mathrm{cost}(P,\mathcal{A}))} \right| \right] \\
\leq \quad & 4 \sqrt{\gamma \cdot k\sqrt{k} \cdot \log \|P\|_0 \cdot i \cdot \log^3 \varepsilon^{-1} \cdot \frac{1}{|\Omega|}} \\
& + 2 \sqrt{\gamma \cdot k^3 \cdot \log \|P\|_0 \cdot \min(k_i, 2^i) \cdot i \cdot \log^3 \varepsilon^{-1} \cdot \frac{1}{|\Omega|} \cdot \frac{1}{k^2}} \\
\leq \quad & \sqrt{\gamma \cdot k\sqrt{k} \cdot \log \|P\|_0 \cdot i \cdot \log^3 \varepsilon^{-1} \cdot \frac{1}{|\Omega|}} \leq O(1) \frac{\varepsilon^{-2}}{\log^3 \varepsilon^{-1}}.
\end{aligned}
\tag{12}
$$

## 4 Disclosure of Funding Acknowledgements

Kapser Green Larsen was partially supported by the Independent Research Fund Denmark (DFF) under a Sapere Aude Research Leader grant No 9064-00068B.

David Saulpic has received funding from the European Research Council (ERC) under the European Union's Horizon 2020 research and innovation programme (Grant agreement No. 101019564 "The Design of Modern Fully Dynamic Data Structures (MoDynStruct)". 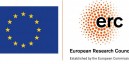

Chris Schwiegelshohn was partially supported by the Independent Research Fund Denmark (DFF) under a Sapere Aude Research Leader grant No 1051-00106B and the Innovation Fund Denmark under grant agreement No 0153-00233A.

Omar Ali Sheikh-Omar was partially supported by the Innovation Fund Denmark under grant agreement No 0153-00233A.

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
