## A  Analysis Details

### A.1  Omitted Proofs (Details for Lemma 3)

**Clustering Nets**   Next, we give a detailed proof of Lemma 4.

*Proof of Lemma 4.* Our objective is to generate a small set of cost vectors that satisfy the desired guarantee.

We first define the cost vectors (the reader familiar with the proof sketch from the main body of the submission may skip this and the next paragraph). For each subset $U$ of size $O(\min(\alpha^{-2}2^i, \alpha^{-2}+k_i)$, we consider the the subspace $\Pi_U$ spanned by $U$. In this subspace we consider $(\alpha/2^i) \cdot \sqrt{\text{cost}(p, \mathcal{A})}$-nets of every ball centered around $\Pi_U p$ with radius $60 \cdot 2^i/2 \cdot \sqrt{\text{cost}(p, \mathcal{A})}$ for all $p \in P$. Such a net has size $\exp(\gamma \cdot \textbf{rank}(U)i \log \alpha)$, for some constant $\gamma$ and there exist at most $\|P\|_0 \cdot \exp(\gamma \cdot |U|i \log \alpha)$ many such nets. Furthermore, there are at most $\binom{\|P\|_0}{|U|} = \|P\|_0^{|U|}$ such subsets.

Now, for every point $p$, define an exponential sequence $\alpha^2(1 + \alpha/2^i)^j$ for $j \in \{0, \ldots \log 102^i\}$. There exist at most $\|P\|_0$ such sequences and every such sequence consists of at most $O(\alpha^{-1} \cdot 2^i \cdot i)$ many values. We combine every net point in ever ball of every subspace with all values in the exponential sequence to obtain the evaluation for a single candidate center. The overall number of candidate centers is therefore of the order $\|P\|_0^{|U|} \cdot \exp(\gamma \cdot |U|i \log \alpha)$, for a sufficiently large $\gamma$. The overall number of candidate cost vectors is now the number of $k$ subsets of candidate centers, i.e. $\|P\|_0^{k \cdot |U|} \cdot \exp(\gamma \cdot k \cdot |U|i \log \alpha)$. Combined with the bounds on $U$, this yields the desired size. What remains to be shown is that the thus constructed cost vectors satisfy the guarantee of a $(\alpha, k)$-clustering net.

We start by noting that since all points are from clusters of type $i$ with $i \geq 3$, we have

$$\text{cost}(p, \mathcal{A}) \cdot 2^i \leq \text{cost}(p, \mathcal{S}) \leq O(1) \cdot 2^i \text{cost}(p, \mathcal{A})$$

as a straightforward application of the triangle inequality. To simplify the calculations, we use the bound $\text{cost}(p, \mathcal{S}) \leq 2^i \text{cost}(p, \mathcal{A})$; the result then follows by folding in the constant factor in the definition of $\alpha$.

Let $\mathcal{S}$ be a candidate solution and let $\mathcal{S}'$ be the subset of $\mathcal{S}$ serving the points of $P$. For a center $s \in \mathcal{S}'$, let $U_1$ be the subset of vectors of $P$ given by Lemma 5 with $\varepsilon := \alpha \cdot 2^{i/2}$ and let $U_2$ be the subset of vectors of $P$ given by Lemma 5 with $\varepsilon := \alpha$ and let $U_2' = U_2 \cup \mathcal{A}$. Let $U = \begin{cases} U_1 & \text{if } rank(U_1) \leq rank(U_2') \\ U_2' & \text{else} \end{cases}$ and let $\Pi_U$ be the orthogonal matrix of the span of $U$.

Suppose, for every point $p$ in the span of $U$, we are given an $\alpha/(4 \cdot 2^i) \cdot \sqrt{\text{cost}(p, \mathcal{A})}$-net of the Euclidean ball centered around $\Pi_U p$ with radius $60 \cdot 2^i/2 \cdot \sqrt{\text{cost}(p, \mathcal{A})}$. Such a net has size $\exp(\gamma \cdot \textbf{rank}(U)i \log \alpha)$, for some constant $\gamma$ and there exist at most $\|P\|_0 \cdot \exp(\gamma \cdot \textbf{rank}(S)i \log \alpha)$ many such nets. Denote by $s'$ the point in these nets closest to $\Pi_U s$. We now argue that either the cost is preserved for any given point, or that the net determines that this point costs significantly more than than a factor $2^i \text{cost}(p, \mathcal{A})$. Specifically, we argue that

$$\|\Pi_U p - s'\|^2 - \|\Pi_U(p-s)\|^2 \quad \leq \quad \alpha \text{cost}(p, \mathcal{A}) \text{ if } \|\Pi_U(p-s)\| \leq 60 \cdot 2^i/2\sqrt{\text{cost}(p, \mathcal{A})} \quad (13)$$
$$\|\Pi_U p - s'\|^2 \quad \geq \quad 100 \cdot 2^i \cdot \text{cost}(p, \mathcal{A}) \text{ if } \|\Pi_U(p-s)\| \geq 20 \cdot 2^i/2\sqrt{\text{cost}(p, \mathcal{A})}.$$

Let $p' \in P$ be the point with minimal $\text{cost}(p, \mathcal{A})$ satisfying the first condition. Then for any point $p$ that also satisfies this condition $\|\Pi_U s - s'\| \leq \frac{\alpha}{4} \cdot 2^{-i} 2^{i/2} \cdot \sqrt{\text{cost}(p', \mathcal{A})} \leq \frac{\alpha}{4} \cdot 2^{-i/2} \sqrt{\text{cost}(p', \mathcal{A})} \leq \frac{\alpha}{4} \cdot 2^{-i/2} \sqrt{\text{cost}(p, \mathcal{A})}$. We then have

$$\left| \|\Pi_U(p-s)\|^2 - \|\Pi_U p - s'\|^2 \right|$$
$$\leq \quad 2\|\Pi_U(p-s)\| \cdot \|\Pi_U s - s'\|$$
$$\leq \quad 2\|\Pi_U(p-s)\| \cdot \frac{\alpha}{4} \cdot 2^{-i/2} \sqrt{\text{cost}(p, \mathcal{A})} \leq \alpha \text{cost}(p, \mathcal{A})$$

Conversely, suppose $p$ satisfies the second condition. Then $\|\Pi_U(p-s)\| \geq 20 \cdot 2^i/2\sqrt{\text{cost}(p, \mathcal{A})}$. Let $q$ be the point in the $(\alpha/2^i)$-net of the ball centered around $\Pi_U p$ with radius $20 \cdot 2^i$ closest to $\Pi_U s$. This implies $\|q - \Pi_U p\| \geq 20 \cdot 2^i/2 \cdot \sqrt{\text{cost}(p, \mathcal{A})} - (\alpha/2^i) \cdot \sqrt{\text{cost}(p, \mathcal{A})} \geq 10 \cdot 2^i/2 \cdot \sqrt{\text{cost}(p, \mathcal{A})}$.

Next, we consider a net over all values $\|(I - \Pi_U \Pi_U^T)s\|$. For every point $p$, define an exponential sequence $\alpha \cdot 2^{-i/2}(1 + \alpha/(4 \cdot 2^i))^j \cdot \sqrt{\cost(p, \mathcal{A})}$ for $j \in \{0, \dots \log(10\alpha^{-2} \cdot 2^i) \cdot \frac{8 \cdot 2^i}{\alpha}\}$. There exist at most $\|P\|_0$ such sequences and every such sequence consists of at most $O(\alpha^{-1} \cdot 2^i \cdot i \log \alpha)$ many values. Similar to the bound above, we will show that there exists an $s''$ in the union of sequences such that

$$|s''^2 - \|(I - \Pi_U \Pi_U^T)s\|^2| \quad \leq \quad \alpha \cdot \cost(p, \mathcal{A}) \tag{14}$$

Observe that the point $p'$ closest to $s$ is contained in $U$. This implies for all $p \in P$ that $\|(I - \Pi_U \Pi_U^T)s\| \leq \sqrt{\cost(p', s)} \leq \sqrt{\cost(p, s)}$. Now, any two successive elements of the sequence have a ratio of $(1 + \alpha/(4 \cdot 2^i))$. Hence, we have $|s'' - \|(I - \Pi_U \Pi_U^T)s\|| \leq \|(I - \Pi_U \Pi_U^T)s\| \cdot \alpha/(4 \cdot 2^i) \leq \alpha\sqrt{\cost(p', \mathcal{A})} \leq \alpha/(4 \cdot 2^i) \leq \frac{\alpha}{4} \cdot 2^{-i/2}\sqrt{\cost(p, \mathcal{A})}$. Thus

$$\left| s''^2 - \|(I - \Pi_U \Pi_U^T)s\|^2 \right|$$
$$\leq \quad 2\|(I - \Pi_U \Pi_U^T)s\| \cdot \left| s'' - \|(I - \Pi_U \Pi_U^T)s\| \right|$$
$$\leq \quad 2\|(I - \Pi_U \Pi_U^T)s\| \cdot \frac{\alpha}{4} \cdot 2^{-i/2}\sqrt{\cost(p, \mathcal{A})} \leq \alpha\cost(p, \mathcal{A})$$

We now conclude for a single center $s$. Assume that the point in $P$ closest to $s$ is included in $U$. Let $s'$ be the point in the span of $U$ satisfying Eq. 13 and let $s''$ be the element in the exponential sequence satisfying Eq. 14. We wish to show that for some absolute constant $\beta$

$$\|\|p - s\|^2 \quad - \quad \left( \|\Pi_U p - s'\|^2 + \|(I - \Pi_U \Pi_U^T)p\|^2 + s'' \right) |$$
$$\leq \quad \beta \cdot \alpha \cdot \sqrt{\cost(p, \mathcal{A})\|p - s\|^2} \text{ if } \|p - s\| \leq 60 \cdot 2^{i/2}\sqrt{\cost(p, \mathcal{A})} \tag{15}$$
$$\|\|p - s\|^2 \quad - \quad \left( \|\Pi_U p - s'\|^2 + \|(I - \Pi_U \Pi_U^T)p\|^2 + s'' \right) |$$
$$\geq \quad 10 \cdot 2^i \cdot \cost(p, \mathcal{A}) \text{ if } \|p - s\| \geq 60 \cdot 2^{i/2}\sqrt{\cost(p, \mathcal{A})}$$

First, we decompose $\|p - s\|^2$ as follows. We have

$$\|p - s\|^2 \quad = \quad \|\Pi_U(p - s)\|^2 + \|(I - \Pi_U \Pi_U^T)(p - s)\|^2$$
$$\leq \quad \|\Pi_U(p - s)\|^2 + \|(I - \Pi_U \Pi_U^T)p\|^2 + \|(I - \Pi_U \Pi_U^T)s\|^2 - 2p^T(I - \Pi_U \Pi_U^T)s$$

We first focus on the case that $\|p - s\| \leq 60 \cdot 2^i/2\sqrt{\cost(p, \mathcal{A})}$. Since projection only decreases the norm, this implies that $\|\Pi_U(p - s)\|$ and $\|(I - \Pi_U \Pi_U^T)(p - s)\|$ are less than $60 \cdot 2^i/2\sqrt{\cost(p, \mathcal{A})}$ as well. This implies that Equations 13 and 15 hold, which implies

$$\|\|p - s\|^2 - \left( \|\Pi_U p - s'\|^2 + \|(I - \Pi_U \Pi_U^T)p\|^2 + s'' \right) | \leq \alpha \cdot \cost(p, \mathcal{A}) + |2p^T(I - \Pi_U \Pi_U^T)s|$$

Since the point closest to $s$ is contained in $U$, we have $\|(I - \Pi_U \Pi_U^T)s\| \leq \sqrt{\cost(p, s)}$. Suppose $|U| = \alpha^{-2} \cdot 2^i$. Then Lemma 5 gives

$$\|(I - \Pi_U \Pi_U^T)p\|^2 + \|(I - \Pi_U \Pi_U^T)s\|^2$$
$$\leq \quad \|p - s\|^2 + |2p^T(I - \Pi_U \Pi_U^T)s|$$
$$\leq \quad \|p - s\|^2 + 2\alpha \cdot 2^{-i/2}\|(I - \Pi_U \Pi_U^T)p\| \min_{p \in P} \|p - s\|.$$

This implies

$$\|(I - \Pi_U \Pi_U^T)p\| \leq 2\|p - s\| \leq 4 \cdot 2^{i/2}\sqrt{\cost(p, \mathcal{A})},$$

and hence

$$p^T(I - \Pi_U \Pi_U^T)s \leq \alpha \cdot 2^{-i/2}\|(I - \Pi_U \Pi_U^T)p\| \min_{p \in P} \|p - s\| \leq 2\alpha\sqrt{\cost(p, \mathcal{A})\cost(p, s)}.$$

as desired. Now suppose $\alpha^{-2} + k_i$. In this case

$$\|(I - \Pi_U \Pi_U^T)p\| \leq \sqrt{\cost(p, \mathcal{A})}.$$

Consequently,

$$p^T(I - \Pi_U \Pi_U^T)s \leq \alpha \cdot \|(I - \Pi_U \Pi_U^T)p\| \min_{p \in P} \|p - s\| \leq 2\alpha\sqrt{\cost(p, \mathcal{A})\cost(p, s)}$$

as desired.

We now focus on the case that $\|p - s\| \geq 60 \cdot 2^{i/2}\sqrt{\text{cost}(p, \mathcal{A})}$. In this case, either $\|\Pi_U(p - s)\|$ or $\|(I - \Pi_U\Pi_U^T)(p - s)\|$ are at least $40 \cdot 2^i/2\sqrt{\text{cost}(p, \mathcal{A})}$. In the former case, Eq. 13 shows that we rule out $s$ as a viable center for $p$. In the latter case, we have due to Lemma 5 $\|(I - \Pi_U\Pi_U^T)(p - s)\|2 = \|(I - \Pi_U\Pi_U^T)p\|^2 + \|(I - \Pi_U\Pi_U^T)s\|^2 - 2p^T(I - \Pi_U\Pi_U^T)s \geq (1-\alpha)\cdot\left(\|(I - \Pi_U\Pi_U^T)p\|^2 + \|(I - \Pi_U\Pi_U^T)s\|^2\right)$. Here, either $\|(I-\Pi_U\Pi_U^T)p\|$ or $\|(I-\Pi_U\Pi_U^T)s\|$ are now at least $20 \cdot 2^i/2\sqrt{\text{cost}(p, \mathcal{A})}$. But the latter cannot be true, as $\|(I - \Pi_U\Pi_U^T)s\|$ is less than the distance of $s$ to its closest point. Therefore simply evaluating $\|(I - \Pi_U\Pi_U^T)p\|$ rules out $s$ as a viable center.

We can now conclude overall. By assumption, all points were members of clusters of type $i$. Therefore, there exists at least one viable center for every point. Thus only the upper equation in 15 applies. The overall lemma now follows by rescaling $\alpha$. □

We also require an additional net that works for low dimensions.

**Lemma 9** (Compare Lemma 22 of Cohen-Addad et al. [2022])**.** *Let $P$ be a set of points in $d$ dimensional Euclidean space, $k$ a positive integer and $\mathcal{A}$ be a candidate solution. Define $\mathbb{C}$ to be the set of possible candidate centers such that the clusters induced by $\mathcal{A}$ are of type $i$, with $3 \leq i \leq \log 1/\varepsilon^2$. For all $\alpha \leq 1/2$, there exists an $(\alpha, k)$-clustering net $\mathbb{N}$ of $\mathbb{C}$ with*

$$|\mathbb{N}| \leq \exp\left(\gamma \cdot k \cdot d \cdot i \log(4/\alpha)\right),$$

*where $\gamma$ is an absolute constant.*

*Proof.* The only difference to Lemma 22 of Cohen-Addad et al. [2022] is that the nets are required to have an error of $\alpha \cdot \sqrt{\text{cost}(p, \mathcal{S})\text{cost}(p, \mathcal{A})}$ rather than $\alpha \cdot (\text{cost}(p, \mathcal{S}) + \text{cost}(p, \mathcal{A}))$. This can be done by rescaling $\varepsilon$ by $2^{-i}$, which in turn is absorbed by the constant $\gamma$ as $2^i \leq O(1) \cdot \varepsilon^{-2}$. □

**Variance Bounds**

**Controlling Eq. 8** Here, we use Lemma 9 and Assumption 2 to show that the number of cost vectors in $\mathbb{N}_{h+1} \times \mathbb{N}_h$ is at most $\exp\left(\gamma \cdot k \cdot \log\|P\|_0 \cdot \varepsilon^{-2}\log h/\varepsilon\right)$. Conditioned on event $\mathcal{E}$, we therefore have

$$
\sum_{\log \varepsilon^{-2}}^{\infty} \mathbb{E}_{\Omega}\mathbb{E}_g\left[\sup_{v^{\mathcal{S},h+1}-v^{\mathcal{S},h}\in\mathbb{N}_{h+1}\times\mathbb{N}_h}\left|\frac{\sum_{C_j\in T_i}\sum_{p\in C_j\cap\Omega}(v_p^{\mathcal{S},2^{-(h+1)}} - v_p^{\mathcal{S},2^{-h}})w_p}{(\text{cost}(P,\mathcal{S}) + \text{cost}(P,\mathcal{A}))}g_p\right|\Bigg|\mathcal{E}\right]
$$

$$
\leq \sum_{\log \varepsilon^{-2}}^{\infty} O(1) \cdot \left(\sqrt{\gamma \cdot k \cdot \log\|P\|_0 \cdot \varepsilon^{-2} \cdot \log h/\varepsilon \cdot \frac{2^{-2h}}{|\Omega|}}\right.
$$

$$
\leq \sum_1^{\infty} O(1) \cdot \sqrt{\gamma \cdot k \cdot \log\|P\|_0 \cdot \log h/\varepsilon \cdot \frac{2^{-2h}}{|\Omega|}}
$$

$$
\leq O(1) \cdot \sqrt{\gamma \cdot k \cdot \log\|P\|_0 \cdot \log 1/\varepsilon \cdot \frac{1}{|\Omega|}}
$$

Similarly, if $\mathcal{E}$ does not hold, we have

$$\sum_{\log \varepsilon^{-2}}^{\infty} \mathbb{E}_\Omega \mathbb{E}_g \left[ \sup_{v^{\mathcal{S},h+1}-v^{\mathcal{S},h}\in \mathbb{N}_{h+1}\times\mathbb{N}_h} \left| \frac{\sum_{C_j\in T_i}\sum_{p\in C_j\cap\Omega}(v_p^{\mathcal{S},2^{-(h+1)}}-v_p^{\mathcal{S},2^{-h}})w_p}{(\text{cost}(P,\mathcal{S})+\text{cost}(P,\mathcal{A}))}g_p \right| \middle| \overline{\mathcal{E}} \right]$$

$$\leq \sum_{\log \varepsilon^{-2}}^{\infty} O(1)\cdot\sqrt{\gamma\cdot k\cdot\log\|P\|_0\cdot\varepsilon^{-2}\cdot\log h/\varepsilon\cdot\frac{2^{-2h}k}{|\Omega|}}$$

$$\leq \sum_{1}^{\infty} O(1)\cdot\sqrt{\gamma\cdot k\cdot\log\|P\|_0\cdot\log h/\varepsilon\cdot\frac{2^{-2h}k}{|\Omega|}}$$

$$\leq O(1)\cdot\sqrt{\gamma\cdot k\cdot\log\|P\|_0\cdot\log 1/\varepsilon\cdot\frac{k}{|\Omega|}}$$

We have $\mathbb{P}[\overline{\mathcal{E}}] \leq 1/k^2$ due to Lemma 6. Since $\|P\|_0 \leq \text{poly}(k,\varepsilon^{-1})$, $2^i \leq O(1)\cdot\varepsilon^{-2}$ and by our choice of $|\Omega|$, we can combine the last two equations with the law of total expectation to obtain

$$\sum_{\log \varepsilon^{-2}}^{\infty} \mathbb{E}_\Omega \mathbb{E}_g \left[ \sup_{v^{\mathcal{S},h+1}-v^{\mathcal{S},h}\in \mathbb{N}_{h+1}\times\mathbb{N}_h} \left| \frac{\sum_{C_j\in T_i}\sum_{p\in C_j\cap\Omega}(v_p^{\mathcal{S},2^{-(h+1)}}-v_p^{\mathcal{S},2^{-h}})w_p}{(\text{cost}(P,\mathcal{S})+\text{cost}(P,\mathcal{A}))}g_p \right| \right]$$

$$\leq O(1)\cdot\sqrt{k\cdot\log k\cdot\sqrt{k}\cdot\log^5\varepsilon^{-1}\cdot\frac{1}{|\Omega|}}. \tag{16}$$

Observe that Eq. 16 and Eq. 12 are essentially identical up to lower order terms.

**Controlling Eq. 6**  Here, we first split Eq. 6 into two estimators that will be easier to handle. We split the estimator into two parts as follows. First, let $q_j := \frac{\sum_{p\in C_j}v_p^{\mathcal{S},2^{-1}}}{|C_j|}$. Now we consider

$$\frac{1}{|\Omega|}\frac{\sum_{C_j\in T_i}\sum_{p\in C_j\cap\Omega}(v_p^{\mathcal{S},2^{-1}}-q_j)w_p}{(\text{cost}(P,\mathcal{S})+\text{cost}(P,\mathcal{A}))}g_p \tag{17}$$

$$+\frac{1}{|\Omega|}\frac{\sum_{C_j\in T_i}\sum_{p\in C_j\cap\Omega}q_j\cdot w_p}{(\text{cost}(P,\mathcal{S})+\text{cost}(P,\mathcal{A}))}g_p \tag{18}$$

Thus, Equation 6 becomes

$$\mathbb{E}_\Omega \mathbb{E}_g \left[ \sup_{v^{\mathcal{S},1}\in\mathbb{N}_{2^{-1}}} \left| \frac{\sum_{C_j\in T_i}\sum_{p\in C_j\cap\Omega}v_p^{\mathcal{S},2^{-1}}w_p}{|\Omega|\cdot(\text{cost}(P,\mathcal{S})+\text{cost}(P,\mathcal{A}))}g_p \right| \right]$$

$$= \mathbb{E}_\Omega \mathbb{E}_g \left[ \sup_{v^{\mathcal{S},1}\in\mathbb{N}_{2^{-1}}} \left| \frac{\sum_{C_j\in T_i}\sum_{p\in C_j\cap\Omega}|v_p^{\mathcal{S},2^{-1}}-q_j|w_p}{|\Omega|\cdot(\text{cost}(P,\mathcal{S})+\text{cost}(P,\mathcal{A}))}g_p \right| \right] \tag{19}$$

$$+ \mathbb{E}_\Omega \mathbb{E}_g \left[ \sup_{v^{\mathcal{S},1}\in\mathbb{N}_{2^{-1}}} \left| \frac{\sum_{C_j\in T_i}\sum_{p\in C_j\cap\Omega}q_j\cdot w_p}{|\Omega|\cdot(\text{cost}(P,\mathcal{S})+\text{cost}(P,\mathcal{A}))}g_p \right| \right] \tag{20}$$

Due to Assumption 4, we have $\text{cost}(T_i,\mathcal{S}) = O(1)\cdot k_i\text{cost}(C_j,\mathcal{S})$, for any $C_j\in T_i$ Thus

$$\mathbb{E}_\Omega \mathbb{E}_g \left[ \sup_{v^{\mathcal{S},1}\in\mathbb{N}_{2^{-1}}} \left| \frac{\sum_{C_j\in T_i}\sum_{p\in C_j\cap\Omega}q_j\cdot w_p}{|\Omega|\cdot(\text{cost}(P,\mathcal{S})+\text{cost}(P,\mathcal{A}))}g_p \right| \right]$$

$$\leq \mathbb{E}_\Omega \mathbb{E}_g \left[ \sup_{v^{\mathcal{S},1}\in\mathbb{N}_{2^{-1}}} \left| \frac{\sum_{C_j\in T_i}\sum_{p\in C_j\cap\Omega}q_j\cdot w_p}{|\Omega|\cdot\text{cost}(T_i,\mathcal{S})}g_p \right| \right]$$

$$\leq \mathbb{E}_\Omega \mathbb{E}_g \left[ \sup_{v^{\mathcal{S},1}\in\mathbb{N}_{2^{-1}}} \left| \max_{C_j\in T_i}\frac{\sum_{p\in C_j\cap\Omega}q_j\cdot w_p}{|\Omega|\cdot\text{cost}(C_j,\mathcal{S})}g_p \right| \right] \tag{21}$$

For the variance of the estimator used for Eq. 19, we use the following lemma.

**Lemma 10.** *If Assumption 4 holds, the variance of* $\frac{1}{|\Omega|}\frac{\sum_{C_j\in T_i}\sum_{p\in C_j\cap\Omega}(v_p^{\mathcal{S},2^{-1}}-q_j)w_p}{(cost(P,\mathcal{S})+cost(P,\mathcal{A}))}g_p$ *is at most*

$$\gamma\cdot\frac{1}{|\Omega|}\cdot\frac{k\cdot k_i\cdot 2^i}{(k+k_i\cdot 2^i)^2}\qquad\text{conditioned on event }\mathcal{E}$$

$$\gamma\cdot\frac{k}{|\Omega|}\cdot\frac{k\cdot k_i\cdot 2^i}{(k+k_i\cdot 2^i)^2}\qquad\text{conditioned on event }\overline{\mathcal{E}}$$

*for an absolute constant* $\gamma$.

*Proof.* We will bound $|v_p^{\mathcal{S},2^{-1}}-q_j|$ for any point $p\in C_j$. Due to the triangle inequality and by Assumption 4 which states that all points have roughly equal distance to their center in $\mathcal{A}$, we have

$$\left|\sqrt{v_p^{\mathcal{S},2^{-1}}}-\sqrt{q_j}\right|\le O(1)\cdot\sqrt{\text{cost}(p,\mathcal{A})}.$$

Futhermore, again due to the triangle inequality, $C_j\in T_i$ with $i>3$ and Assumption 4, we have $\left(\sqrt{v_p^{\mathcal{S},2^{-1}}}+\sqrt{q_j}\right)=O(1)\sqrt{\text{cost}(p,\mathcal{S})}$. Therefore

$$|v_p^{\mathcal{S},2^{-1}}-q_j|\;=\;\left|\sqrt{v_p^{\mathcal{S},2^{-1}}}-\sqrt{q_j}\right|\cdot\left(\sqrt{v_p^{\mathcal{S},2^{-1}}}+\sqrt{q_j}\right)=O(1)\sqrt{\text{cost}(p,\mathcal{S})\text{cost}(p,\mathcal{A})}$$

Using this bound and the same steps as in Lemma 7, we then have

$$\sum_{C_j\in T_i}\sum_{p\in C_j\cap\Omega}\frac{1}{|\Omega|^2}\left(\frac{(v_p^{\mathcal{S},2^{-1}}-q_j)w_p}{(\text{cost}(P,\mathcal{S})+\text{cost}(P,\mathcal{A}))}\right)^2$$

$$\le\;O(1)\cdot\sum_{C_j\in T_i}\sum_{p\in C_j\cap\Omega}\frac{1}{|\Omega|^2}\frac{\text{cost}(p,\mathcal{S})\text{cost}(p,\mathcal{A})\text{cost}(P,\mathcal{S})^2|C_j|^2}{\text{cost}(C_j,\mathcal{A})^2\cdot(\text{cost}(P,\mathcal{S})+\text{cost}(P,\mathcal{A}))^2}$$

$$\le\;O(1)\cdot\sum_{C_j\in T_i}\sum_{p\in C_j\cap\Omega}\left(\frac{k}{k_i\cdot|\Omega|^2}\right)\cdot\frac{k\cdot k_i\cdot 2^i}{(k+k_i\cdot 2^i)^2}$$

Conditioned on event $\mathcal{E}$, this now becomes $O(1)\cdot\frac{1}{|\Omega|}\cdot\frac{k\cdot k_i\cdot 2^i}{(k+k_i\cdot 2^i)^2}$ and similarly, if event $\mathcal{E}$ does not hold, we have the bound $\frac{k}{|\Omega|}\cdot\frac{k\cdot k_i\cdot 2^i}{(k+k_i\cdot 2^i)^2}$. $\qquad\square$

We now focus on the variance of the estimator used for Eq. 20. Due to Assumption 4, we have $\text{cost}(T_i,\mathcal{S})=O(1)\cdot k_i\text{cost}(C_j,\mathcal{S})$, for any $C_j\in T_i$ Thus

$$\mathbb{E}_\Omega\mathbb{E}_g\left[\left\|\sup_{v^{\mathcal{S},1}\in\mathbb{N}_{2^{-1}}}\frac{\sum_{C_j\in T_i}\sum_{p\in C_j\cap\Omega}q_j\cdot w_p}{|\Omega|\cdot(\text{cost}(P,\mathcal{S})+\text{cost}(P,\mathcal{A}))}g_p\right\|\right]$$

$$\le\;\mathbb{E}_\Omega\mathbb{E}_g\left[\left\|\sup_{v^{\mathcal{S},1}\in\mathbb{N}_{2^{-1}}}\frac{\sum_{C_j\in T_i}\sum_{p\in C_j\cap\Omega}q_j\cdot w_p}{|\Omega|\cdot\text{cost}(T_i,\mathcal{S})}g_p\right\|\right]$$

$$\le\;\mathbb{E}_\Omega\mathbb{E}_g\left[\left\|\sup_{v^{\mathcal{S},1}\in\mathbb{N}_{2^{-1}}}\max_{C_j\in T_i}\frac{\sum_{p\in C_j\cap\Omega}q_j\cdot w_p}{|\Omega|\cdot\text{cost}(C_j,\mathcal{S})}g_p\right\|\right]\qquad(22)$$

We now obtain the following variance for the estimator used in Equation 22.

**Lemma 11.** *If Assumption 4 holds, the variance of* $\frac{\sum_{p\in C_j\cap\Omega}q_j\cdot w_p}{|\Omega|\cdot cost(C_j,\mathcal{S})}g_p$, *given that* $C_j\in T_i$ *with* $i\in\{3,\ldots,\log\varepsilon^{-2}\}$ *is at most*

$$\gamma\cdot\frac{k}{|\Omega|}\qquad\text{conditioned on event }\mathcal{E}$$

$$\gamma\cdot\frac{k^2}{|\Omega|}\qquad\text{conditioned on event }\overline{\mathcal{E}}$$

*for an absolute constant* $\gamma$.

*Proof.* Recall by Assumption 4 $\text{cost}(P, \mathcal{S}) = O(1) \cdot k \cdot \text{cost}(C_j, \mathcal{S})$. We have

$$\sum_{p \in C_j \cap \Omega} \left( \frac{q_j \cdot w_p}{|\Omega| \cdot \text{cost}(C_j, \mathcal{S})} \right)^2$$

$$= \sum_{p \in C_j \cap \Omega} \left( \frac{q_j \cdot \text{cost}(P, \mathcal{A}) \cdot |C_j|}{|\Omega| \cdot \text{cost}(C_j, \mathcal{A}) \cdot \text{cost}(C_j, \mathcal{S})} \right)^2$$

$$= O(1) \cdot \sum_{p \in C_j \cap \Omega} \left( \frac{k}{|\Omega|} \right)^2$$

Conditioned on event $\mathcal{E}$, $|C_j \cap \Omega| = \frac{1}{k} \cdot |\Omega|$ and this now becomes $O(1) \cdot \frac{k}{|\Omega|}$. Otherwise, we have the bound $\frac{k^2}{|\Omega|}$. $\qquad\square$

We now bound Equations 19 and 22. For the former, we have $|\mathbb{N}_{2^{-1}}| \leq \exp\left(\gamma \cdot k \cdot \log \|P\|_0 \cdot \min(k_i, 2^i) \cdot i\right)$. As was the case in the proof of Eq. 7), we split the vectors in the nets by $k_i \leq \sqrt{k}$ or by $k_i \geq \sqrt{k}$. In both cases, we have $\min(k_i, 2^i) \frac{k \cdot k_i \cdot 2^i}{(k + k_i \cdot 2^i)^2} \leq \sqrt{k}$. Thus, combined with Lemma 10 and conditioning on event $\mathcal{E}$, we have

$$\mathbb{E}_\Omega \mathbb{E}_g \left[ \sup_{v^{\mathcal{S},1} \in \mathbb{N}_{2^{-1}}} \left| \frac{\sum_{C_j \in T_i} \sum_{p \in C_j \cap \Omega} |v_p^{\mathcal{S},2^{-1}} - q_j|w_p}{|\Omega| \cdot (\text{cost}(P, \mathcal{S}) + \text{cost}(P, \mathcal{A}))} g_p \right| \, \Big| \, \mathcal{E} \right]$$

$$\leq \mathbb{E}_\Omega \mathbb{E}_g \left[ \sup_{v^{\mathcal{S},1} \in \mathbb{N}_{2^{-1}}, k_i \leq \sqrt{k}} \left| \frac{\sum_{C_j \in T_i} \sum_{p \in C_j \cap \Omega} |v_p^{\mathcal{S},2^{-1}} - q_j|w_p}{|\Omega| \cdot (\text{cost}(P, \mathcal{S}) + \text{cost}(P, \mathcal{A}))} g_p \right| \, \Big| \, \mathcal{E} \right]$$

$$+ \mathbb{E}_\Omega \mathbb{E}_g \left[ \sup_{v^{\mathcal{S},1} \in \mathbb{N}_{2^{-1}}, k_i \geq \sqrt{k}} \left| \frac{\sum_{C_j \in T_i} \sum_{p \in C_j \cap \Omega} |v_p^{\mathcal{S},2^{-1}} - q_j|w_p}{|\Omega| \cdot (\text{cost}(P, \mathcal{S}) + \text{cost}(P, \mathcal{A}))} g_p \right| \, \Big| \, \mathcal{E} \right]$$

$$= O(1) \sqrt{\gamma \cdot k \cdot \log \|P\|_0 \cdot \sqrt{k} \cdot i \cdot \frac{1}{|\Omega|}}$$

Similarly, not conditioning on event $\mathcal{E}$ implies

$$\mathbb{E}_\Omega \mathbb{E}_g \left[ \sup_{v^{\mathcal{S},1} \in \mathbb{N}_{2^{-1}}} \left| \frac{\sum_{C_j \in T_i} \sum_{p \in C_j \cap \Omega} |v_p^{\mathcal{S},2^{-1}} - q_j|w_p}{|\Omega| \cdot (\text{cost}(P, \mathcal{S}) + \text{cost}(P, \mathcal{A}))} g_p \right| \, \Big| \, \overline{\mathcal{E}} \right]$$

$$\leq O(1) \sqrt{\gamma \cdot k^2 \cdot \log \|P\|_0 \cdot \min(k_i, 2^i) \cdot i \cdot \frac{k}{|\Omega|}}$$

We have $\mathbb{P}[\overline{\mathcal{E}}] \leq 1/k^2$ due to Lemma 6. Plugging in $\|P\|_0 \leq \text{poly}(k, \varepsilon^{-1})$ and our choice of $|\Omega|$, we can combine the last two equations with the law of total expectation to obtain

$$\mathbb{E}_\Omega \mathbb{E}_g \left[ \sup_{v^{\mathcal{S},1} \in \mathbb{N}_{2^{-1}}} \left| \frac{\sum_{C_j \in T_i} \sum_{p \in C_j \cap \Omega} |v_p^{\mathcal{S},2^{-1}} - q_j|w_p}{|\Omega| \cdot (\text{cost}(P, \mathcal{S}) + \text{cost}(P, \mathcal{A}))} g_p \right| \right]$$

$$\leq O(1) \cdot \sqrt{k \cdot \log k \cdot \sqrt{k} \cdot \log^5 \varepsilon^{-1} \cdot \frac{1}{|\Omega|}} \qquad (23)$$

For the term in Equation 22, we note that $\frac{q_j \cdot w_p}{\text{cost}(C_j, \mathcal{S})} = \text{cost}(P, \mathcal{A})$. Thus, for every cluster, we have a net of size 1, which means we have an overall net of size $k$. We thus obtain

$$\mathbb{E}_\Omega \mathbb{E}_g \left[ \sup_{v^{\mathcal{S},1} \in \mathbb{N}_{2^{-1}}} \max_{C_j \in T_i} \left| \frac{\sum_{p \in C_j \cap \Omega} q_j \cdot w_p}{|\Omega| \cdot \text{cost}(C_j, \mathcal{S})} g_p \right| \, \Big| \, \mathcal{E} \right]$$

$$\leq O(1) \sqrt{\log k \frac{k}{|\Omega|}}$$

Similarly, conditioning on event $\overline{\mathcal{E}}$ implies

$$\mathbb{E}_\Omega \mathbb{E}_g \left[ \sup_{v^{\mathcal{S},1} \in \mathbb{N}_{2^{-1}}} \max_{C_j \in T_i} \left| \frac{\sum_{p \in C_j \cap \Omega} q_j \cdot w_p}{|\Omega| \cdot \text{cost}(C_j, \mathcal{S})} g_p \right| | \overline{\mathcal{E}} \right]$$

$$\leq O(1) \sqrt{\log k \frac{k^2}{|\Omega|}}$$

Combining both terms, using $\mathbb{P}[\overline{\mathcal{E}}] \leq 1/k^2$ due to Lemma 6 and the law of total expectation, we obtain

$$\mathbb{E}_\Omega \mathbb{E}_g \left[ \sup_{v^{\mathcal{S},1} \in \mathbb{N}_{2^{-1}}} \max_{C_j \in T_i} \left| \frac{\sum_{p \in C_j \cap \Omega} q_j \cdot w_p}{|\Omega| \cdot \text{cost}(C_j, \mathcal{S})} g_p \right| \right] \in O(1) \cdot \sqrt{\frac{k \log k}{|\Omega|}} \qquad (24)$$

Combining the bounds in Equations 12, 16, 23 and 24 for the respective terms in Equations 7, 8, 19 and 22 now yields the claim.

## B $k$-Median

The approach for $k$-median is largely the same. The main difference is in the new accuracy requirement of the clustering nets. In what follows, we let $\text{cost}(p, \mathcal{S}) := \min_{s \in \mathcal{S}} \|p - s\|$, i.e. the cost is the closest distance rather than the squared Euclidean distance. This leads to a small but crucial change in the definition of the types. Specifically, the maximum type $T_i$ now satisfies $\text{cost}(p, \mathcal{S}) \leq O(1) \cdot \varepsilon^{-1} \cdot \text{cost}(p, \mathcal{S})$, i.e. $2^i \leq \varepsilon^{-1}$. Previously, we only had $2^i \leq \varepsilon^{-2}$.

**Definition 2.** *Let $I$ be a metric space, $P$ a set of points, $k$ a positive integer, and let $\alpha > 0$ be a precision parameters and let $\mathcal{A}$ be some solution with at most $k'$ centers. Let $\mathbb{C} \subset I^k$ be a (potentially infinite) set of candidate $k$-clusterings. We say that a set of cost vectors $\mathbb{N} \subset \mathbb{R}^{|P|}$ is an $(\alpha, k)$-means clustering net if for every $\mathcal{S} \in \mathbb{C}$ there exists a vector $v' \in \mathbb{N}$ such that the following condition holds. For all $p \in P$,*

$$|v_p^{\mathcal{S}} - v_p'| \leq \alpha \cdot cost(p, \mathcal{A}).$$

We now adapt Lemma4.

**Lemma 12.** *Let $P$ be a set of points in $d$ dimensional Euclidean space, $k$ a positive integer, $\mathcal{A}$ be a candidate solution with $k_i$ clusters and $\gamma$ an absolute constant. Define $\mathbb{C}$ to be the set of possible candidate centers such that a subset $A_i$ the clusters induced by $\mathcal{A}$ are of type $i$, with $3 \leq i \leq \log 1/\varepsilon^2$. For all $\alpha \leq 1/2$, there exists an $(\alpha, k)$-means clustering net $\mathbb{N}$ of $\mathbb{C}$ with*

$$|\mathbb{N}| \leq \exp\left( \gamma \cdot k \cdot \log \|P\|_0 \cdot \min(k_i + \alpha^{-2}, \alpha^{-2} \cdot 2^{2i}) \cdot i \log \frac{1}{\alpha} \right).$$

*Proof.* The proof is largely identical to that of Lemma 4. We first establish

$$\|p - s\|^2 = \|\Pi(p - s)\|^2 + \|(I - \Pi\Pi^T)p\|^2 + \|(I - \Pi\Pi^T)s\|^2 \pm \varepsilon \cdot \|p - s\| \cdot \text{cost}(p, \mathcal{A}), \quad (25)$$

which requires nets of size

$$|\mathbb{N}| \leq \exp\left( \gamma \cdot k \cdot \log \|P\|_0 \cdot \min(k_i + \alpha^{-2}, \alpha^{-2} \cdot 2^{2i}) \cdot i \log \frac{1}{\alpha} \right).$$

Notice the higher dependency on $2^i$, accounted for due to $\text{cost}(p, \mathcal{S})^2 = 2^{2i}\text{cost}(p, \mathcal{A})^2$. Once we have Equation 25 we can immediately infer

$$\left| \|p - s\| - \sqrt{\|\Pi(p - s)\|^2 + \|(I - \Pi\Pi^T)p\|^2 + \|(I - \Pi\Pi^T)s\|^2} \right|$$

$$\leq \varepsilon \cdot \frac{\|p - s\| \cdot \text{cost}(p, \mathcal{A})}{\|p - s\| + \sqrt{\|\Pi(p - s)\|^2 + \|(I - \Pi\Pi^T)p\|^2 + \|(I - \Pi\Pi^T)s\|^2}} \leq \varepsilon \cdot \text{cost}(p, \mathcal{A}).$$

The remaining parts of the proof are not affected. $\qquad\square$

The remaining calculations are now almost identical to the ones given for $k$-means. The only difference is the new net size and the accuracy of the net. Adapting the variance bound in Lemmas 7, 10, and 11 for $\frac{\sum_{C_j \in T_i} \sum_{p \in C_j \cap \Omega} (v_p^{S,2^{-(h+1)}} - v_p^{S,2^{-h}}) w_p \cdot g_p}{|\Omega| \cdot (\text{cost}(P,S) + \text{cost}(P,A))}$ we now obtain

$$\gamma \cdot \frac{2^{-2h}}{|\Omega|} \cdot \frac{k \cdot k_i}{(k + k_i \cdot 2^i)^2} \qquad \text{conditioned on event } \mathcal{E}$$

$$\gamma \cdot \frac{2^{-2h} \cdot k}{|\Omega|} \cdot \frac{k \cdot k_i}{(k + k_i \cdot 2^i)^2} \qquad \text{conditioned on event } \overline{\mathcal{E}}.$$

Thus, we aim to optimize the term

$$\min(k_i, 2^{2i}) \cdot \frac{k \cdot k_i}{(k + k_i \cdot 2^i)^2}$$

For $k_i \leq k^{2/3}$, we have $k_i \cdot \frac{k \cdot k_i}{k^2} \leq k^{1/3}$. For $k_i \geq k^{2/3}$, we have $2^{2i} \cdot \frac{k \cdot k_i}{k_i^2 2^{2i}} \leq k^{1/3}$. Pluggin this term into the chaining analysis then yields the desired bound of $\sqrt{\gamma \cdot k \cdot k^{1/3} \cdot \frac{1}{|\Omega|}} \cdot \text{polylog}(k, \varepsilon)$. Setting this term to be less than $\frac{\varepsilon}{\log^2 \varepsilon^{-1}}$ yields the desired coreset bound.

## C   Experimental Results

We complement our results with an empirical evaluation of coreset algorithms. Our aim is to only study whether a linear dependency on $k$ is (empirically) the correct answer or not. For a more complete experimental study on coresets, including the sampling distributions studied here, we refer to Schwiegelshohn and Sheikh-Omar [2022]. Specifically, we ran the algorithm used in our analysis (Group Sampling) as well as the popular Sensitivity Sampling (?) and uniform sampling on synthetic and real world data. Here, we mainly focus on the performance of the three real world instances *Census*[4], *Covertype*[5] and *Tower*[6]. It is hard to compute the distortion of a given coreset, so for the real world data sets, we sampled $|\Omega| = 200k$ points from each distribution, followed by running $k$-means++ (see Arthur and Vassilvitskii [2007]) 10 times and compared the costs, as well as the distortion on the found solution (see Fig. 6). Every coreset was computed 10 times as well. The distortion is the maximum distortion over each run among all evaluated solutions.

In terms of cost, all algorithms computed solutions of roughly equal value. In terms of distortion, uniform sampling expectedly had the worst bounds. Somewhat interestingly, the Group Sampling algorithm which currently has a better worst case bound than Sensitivity Sampling tends to have a worse distortion. We leave it to future work to find an analysis of Sensitivity Sampling that matches (or outperforms) Group Sampling. Furthermore, there is a small increase in distortion with increasing $k$. While this suggests that a linear dependency in $k$ is not the correct answer, the increase in distortion is too small to suggest that a $k^{1+\delta}$ dependency for some $\delta > 0$ will be the correct answer. It may also be too difficult to accurately measure distortion on real world instances as uniform sampling, which is known to be an extremely bad coreset algorithm in theory, performs moderately competitively.

**Synthetic Data Sets**   The first synthetic instance, henceforth called *Instance 1*, is simply the hard instance from Cohen-Addad et al. [2022], which are just $m$ standard unit vectors in $\mathbb{R}^m$. Cohen-Addad et al. [2022] showed that for $m \in \Omega(k/\varepsilon^2)$, no significant space savings can be achieved. We varied $m$ for additional values $m = \{k^{1.25}/\varepsilon^2, k^{1.5}/\varepsilon^2\}$ with $\varepsilon \in \{0.2, 0.1, 0.05, 0.01\}$ and $k \in \{10, 20, 50, 70, 100\}$.

The second synthetic instance, henceforth called *Instance 2*, is constructed as follows. We select $k/2$ standard basis vectors. Around each of the standard basis vectors, we sampled $m = \{1/\varepsilon^2, k^{0.25}/\varepsilon^2, k^{0.5}/\varepsilon^2\}$ random vectors with norm $\varepsilon$, with $\varepsilon \in \{0.2, 0.1, 0.05, 0.01\}$ and $k \in \{10, 20, 50, 70, 100\}$. The reason for choosing this data set is because this is the arguably simplest instance for which we could not give an improve the $\tilde{O}(k^{1.5}/\varepsilon^2)$ analysis given in this paper.

For these data sets, Group Sampling and Uniform Sampling are identical, which is why we only use one in the evaluations.

---

[4]`https://archive.ics.uci.edu/ml/datasets/US+Census+Data+(1990)` (see Kohavi [1996])

[5]`https://archive.ics.uci.edu/ml/datasets/covertype` (see Blackard and Dean [1999])

[6]`http://homepages.uni-paderborn.de/frahling/coremeans.html` (see Frahling and Sohler [2006])

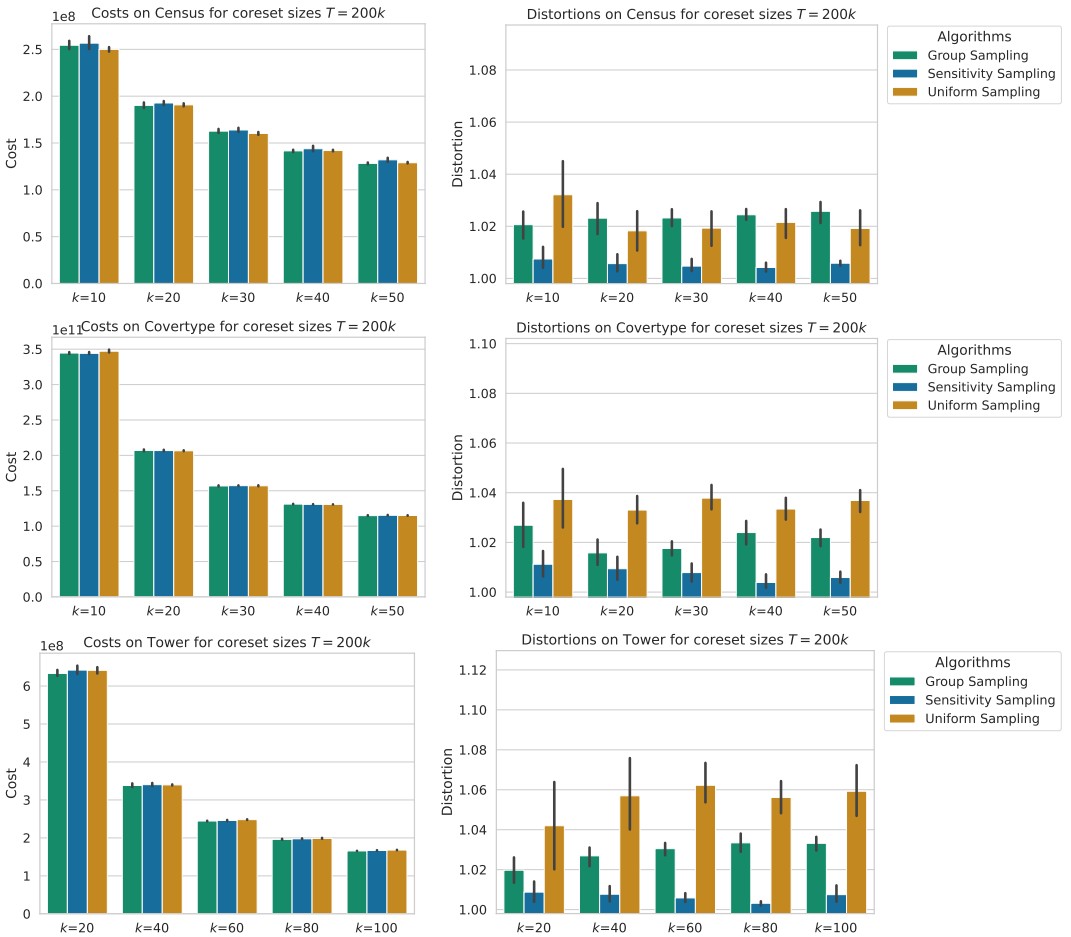

Figure 1: Costs and distortions on real-world data sets. Reported are mean values over the runs, as well as standard deviations indicated by the black bars.

**Evaluations** We selected coresets of sizes proportionate to $k/\varepsilon^2$, $k^{1.25}/\varepsilon^2$ and $k^{1.5}/\varepsilon^2$. For every size, we generated 10 coresets. Subsequently, we generated 100 candidate solutions for every designated coreset, and measured the distortion, i.e. the maximum ratio between coreset cost and original cost (resp. original cost and coreset cost, if the coreset cost became cheaper).

For each instance and every combination of the parameters, we gave a box plot of the distortions and, for completeness, the costs of the algorithm when running $k$-means++ and returning the cheapest one.

**Results** For both synthetic data sets, we observed that no matter how the hard instance was constructed, the distortion measured on the evaluated instances decreased with increasing $k$. Since that was the case even for coresets of size $O(k/\varepsilon^2)$, this suggests that the dependency on $k$ is non-increasing. Between Sensitivity Sampling and Group Sampling/Uniform Sampling, the latter distribution usually obtained slightly better distortions, though overall the two algorithms seem to behave similarly. We conjecture that a both should yield a coreset when sampling $\tilde{O}(k/\varepsilon^2)$ many points.

### C.1 Synthetic Instance 1

**Costs on the Instance 1 ($\epsilon = 0.01$)**

| $k$ | Coreset Size | Sensitivity Sampling | Uniform Sampling |
|---|---|---|---|
| 10 | $k^{1.00}/10\epsilon^2$ | 4.745e+05 (7.093e+02) | 4.745e+05 (6.910e+02) |

| | $k^{1.25}/10\epsilon^2$ | 4.745e+05 (8.286e+02) | 4.745e+05 (7.463e+02) |
|---|---|---|---|
| | $k^{1.50}/10\epsilon^2$ | 4.745e+05 (7.577e+02) | 4.745e+05 (7.057e+02) |
| 20 | $k^{1.00}/10\epsilon^2$ | 1.342e+06 (4.874e+02) | 1.342e+06 (6.252e+02) |
| | $k^{1.25}/10\epsilon^2$ | 1.342e+06 (6.230e+02) | 1.342e+06 (5.781e+02) |
| | $k^{1.50}/10\epsilon^2$ | 1.342e+06 (5.386e+02) | 1.342e+06 (4.889e+02) |
| 50 | $k^{1.00}/10\epsilon^2$ | 5.303e+06 (3.633e+02) | 5.303e+06 (2.869e+02) |
| | $k^{1.25}/10\epsilon^2$ | 5.303e+06 (3.314e+02) | 5.303e+06 (3.352e+02) |
| | $k^{1.50}/10\epsilon^2$ | 5.303e+06 (3.499e+02) | 5.303e+06 (3.202e+02) |
| 70 | $k^{1.00}/10\epsilon^2$ | 8.785e+06 (2.752e+02) | 8.785e+06 (2.729e+02) |
| | $k^{1.25}/10\epsilon^2$ | 8.785e+06 (2.841e+02) | 8.785e+06 (2.570e+02) |
| | $k^{1.50}/10\epsilon^2$ | 8.785e+06 (2.550e+02) | 8.785e+06 (2.361e+02) |
| 100 | $k^{1.00}/10\epsilon^2$ | 1.500e+07 (2.222e+02) | 1.500e+07 (2.139e+02) |
| | $k^{1.25}/10\epsilon^2$ | 1.500e+07 (2.095e+02) | 1.500e+07 (2.456e+02) |
| | $k^{1.50}/10\epsilon^2$ | 1.500e+07 (2.442e+02) | 1.500e+07 (2.230e+02) |

Table 1: Costs of the evaluated algorithms on Instance 1 with $\epsilon = 0.01$. Reported are median values over 10 runs, as well as IQR in paranthesis.

**Costs on the Instance 1 ($\epsilon = 0.05$)**

| $k$ | Coreset Size | Sensitivity Sampling | Uniform Sampling |
|---|---|---|---|
| 10 | $k^{1.00}/10\epsilon^2$ | 1.896e+04 (3.079e+01) | 1.896e+04 (2.783e+01) |
| | $k^{1.25}/10\epsilon^2$ | 1.896e+04 (2.824e+01) | 1.896e+04 (2.732e+01) |
| | $k^{1.50}/10\epsilon^2$ | 1.896e+04 (2.666e+01) | 1.896e+04 (2.891e+01) |
| 20 | $k^{1.00}/10\epsilon^2$ | 5.363e+04 (1.959e+01) | 5.363e+04 (2.021e+01) |
| | $k^{1.25}/10\epsilon^2$ | 5.363e+04 (2.117e+01) | 5.363e+04 (2.121e+01) |
| | $k^{1.50}/10\epsilon^2$ | 5.363e+04 (1.959e+01) | 5.363e+04 (2.060e+01) |
| 50 | $k^{1.00}/10\epsilon^2$ | 2.120e+05 (1.292e+01) | 2.120e+05 (1.313e+01) |
| | $k^{1.25}/10\epsilon^2$ | 2.120e+05 (1.325e+01) | 2.120e+05 (1.282e+01) |
| | $k^{1.50}/10\epsilon^2$ | 2.120e+05 (1.323e+01) | 2.120e+05 (1.363e+01) |
| 70 | $k^{1.00}/10\epsilon^2$ | 3.513e+05 (1.121e+01) | 3.513e+05 (1.102e+01) |
| | $k^{1.25}/10\epsilon^2$ | 3.513e+05 (1.087e+01) | 3.513e+05 (1.137e+01) |
| | $k^{1.50}/10\epsilon^2$ | 3.513e+05 (1.105e+01) | 3.513e+05 (1.174e+01) |
| 100 | $k^{1.00}/10\epsilon^2$ | 5.998e+05 (9.413e+00) | 5.998e+05 (9.632e+00) |
| | $k^{1.25}/10\epsilon^2$ | 5.998e+05 (9.848e+00) | 5.998e+05 (9.281e+00) |
| | $k^{1.50}/10\epsilon^2$ | 5.998e+05 (9.161e+00) | 5.998e+05 (9.166e+00) |

Table 2: Costs of the evaluated algorithms on Instance 1 with $\epsilon = 0.05$. Reported are median values over 100 runs, as well as IQR in paranthesis.

**Costs on the Instance 1 ($\epsilon = 0.1$)**

| $k$ | Coreset Size | Sensitivity Sampling | Uniform Sampling |
|---|---|---|---|
| 10 | $k^{1.00}/10\epsilon^2$ | 4.727e+03 (6.874e+00) | 4.727e+03 (6.884e+00) |
| | $k^{1.25}/10\epsilon^2$ | 4.727e+03 (7.755e+00) | 4.727e+03 (7.798e+00) |
| | $k^{1.50}/10\epsilon^2$ | 4.727e+03 (7.568e+00) | 4.727e+03 (7.365e+00) |
| 20 | $k^{1.00}/10\epsilon^2$ | 1.338e+04 (5.145e+00) | 1.338e+04 (5.519e+00) |
| | $k^{1.25}/10\epsilon^2$ | 1.338e+04 (5.094e+00) | 1.338e+04 (5.478e+00) |
| | $k^{1.50}/10\epsilon^2$ | 1.338e+04 (5.013e+00) | 1.338e+04 (5.624e+00) |
| 50 | $k^{1.00}/10\epsilon^2$ | 5.294e+04 (4.006e+00) | 5.294e+04 (4.065e+00) |
| | $k^{1.25}/10\epsilon^2$ | 5.294e+04 (3.989e+00) | 5.294e+04 (3.890e+00) |

| $k$ | Coreset Size | Sensitivity Sampling | Uniform Sampling |
|---|---|---|---|
|  | $k^{1.50}/10\epsilon^2$ | 5.294e+04 (3.941e+00) | 5.294e+04 (3.915e+00) |
| 70 | $k^{1.00}/10\epsilon^2$ | 8.772e+04 (3.781e+00) | 8.772e+04 (3.822e+00) |
|  | $k^{1.25}/10\epsilon^2$ | 8.772e+04 (3.816e+00) | 8.772e+04 (3.876e+00) |
|  | $k^{1.50}/10\epsilon^2$ | 8.772e+04 (3.828e+00) | 8.772e+04 (3.839e+00) |
| 100 | $k^{1.00}/10\epsilon^2$ | 1.498e+05 (3.988e+00) | 1.498e+05 (3.786e+00) |
|  | $k^{1.25}/10\epsilon^2$ | 1.498e+05 (4.011e+00) | 1.498e+05 (3.943e+00) |
|  | $k^{1.50}/10\epsilon^2$ | 1.498e+05 (4.014e+00) | 1.498e+05 (3.955e+00) |

Table 3: Costs of the evaluated algorithms on Instance 1 with $\epsilon = 0.1$. Reported are median values over 100 runs, as well as IQR in paranthesis.

**Costs on the Instance 1 ($\epsilon = 0.2$)**

| $k$ | Coreset Size | Sensitivity Sampling | Uniform Sampling |
|---|---|---|---|
| 10 | $k^{1.00}/10\epsilon^2$ | 1.168e+03 (2.158e+00) | 1.168e+03 (2.223e+00) |
|  | $k^{1.25}/10\epsilon^2$ | 1.168e+03 (2.153e+00) | 1.168e+03 (2.170e+00) |
|  | $k^{1.50}/10\epsilon^2$ | 1.168e+03 (2.166e+00) | 1.168e+03 (2.095e+00) |
| 20 | $k^{1.00}/10\epsilon^2$ | 3.319e+03 (2.065e+00) | 3.319e+03 (2.096e+00) |
|  | $k^{1.25}/10\epsilon^2$ | 3.319e+03 (2.088e+00) | 3.319e+03 (2.063e+00) |
|  | $k^{1.50}/10\epsilon^2$ | 3.319e+03 (2.077e+00) | 3.319e+03 (2.141e+00) |
| 50 | $k^{1.00}/10\epsilon^2$ | 1.317e+04 (2.446e+00) | 1.317e+04 (2.415e+00) |
|  | $k^{1.25}/10\epsilon^2$ | 1.317e+04 (2.453e+00) | 1.317e+04 (2.379e+00) |
|  | $k^{1.50}/10\epsilon^2$ | 1.317e+04 (2.374e+00) | 1.317e+04 (2.397e+00) |
| 70 | $k^{1.00}/10\epsilon^2$ | 2.184e+04 (2.733e+00) | 2.184e+04 (2.643e+00) |
|  | $k^{1.25}/10\epsilon^2$ | 2.184e+04 (2.730e+00) | 2.184e+04 (2.650e+00) |
|  | $k^{1.50}/10\epsilon^2$ | 2.184e+04 (2.657e+00) | 2.184e+04 (2.727e+00) |
| 100 | $k^{1.00}/10\epsilon^2$ | 3.732e+04 (2.978e+00) | 3.732e+04 (3.016e+00) |
|  | $k^{1.25}/10\epsilon^2$ | 3.732e+04 (2.987e+00) | 3.732e+04 (2.970e+00) |
|  | $k^{1.50}/10\epsilon^2$ | 3.732e+04 (3.058e+00) | 3.732e+04 (3.032e+00) |

Table 4: Costs of the evaluated algorithms on Instance 1 with $\epsilon = 0.2$. Reported are median values over 100 runs, as well as IQR in paranthesis.

**Distortions on the Instance 1 ($\epsilon = 0.01$)**

| $k$ | Coreset Size | Sensitivity Sampling | Uniform Sampling |
|---|---|---|---|
| 10 | $k^{1.00}/10\epsilon^2$ | 1.0000 (0.00000) | 1.0000 (0.00000) |
|  | $k^{1.25}/10\epsilon^2$ | 1.0000 (0.00003) | 1.0000 (0.00003) |
|  | $k^{1.50}/10\epsilon^2$ | 1.0000 (0.00003) | 1.0000 (0.00003) |
| 20 | $k^{1.00}/10\epsilon^2$ | 1.0000 (0.00001) | 1.0000 (0.00001) |
|  | $k^{1.25}/10\epsilon^2$ | 1.0000 (0.00002) | 1.0000 (0.00002) |
|  | $k^{1.50}/10\epsilon^2$ | 1.0000 (0.00001) | 1.0000 (0.00001) |
| 50 | $k^{1.00}/10\epsilon^2$ | 1.0000 (0.00001) | 1.0000 (0.00001) |
|  | $k^{1.25}/10\epsilon^2$ | 1.0000 (0.00001) | 1.0000 (0.00001) |
|  | $k^{1.50}/10\epsilon^2$ | 1.0000 (0.00000) | 1.0000 (0.00000) |
| 70 | $k^{1.00}/10\epsilon^2$ | 1.0000 (0.00001) | 1.0000 (0.00001) |
|  | $k^{1.25}/10\epsilon^2$ | 1.0000 (0.00001) | 1.0000 (0.00001) |
|  | $k^{1.50}/10\epsilon^2$ | 1.0000 (0.00000) | 1.0000 (0.00000) |
| 100 | $k^{1.00}/10\epsilon^2$ | 1.0000 (0.00001) | 1.0000 (0.00001) |
|  | $k^{1.25}/10\epsilon^2$ | 1.0000 (0.00000) | 1.0000 (0.00000) |

| | $k^{1.50}/10\epsilon^2$ | 1.0000 (0.00000) | 1.0000 (0.00000) |

Table 5: Distortions of the evaluated algorithms on Instance 1 with $\epsilon = 0.01$. Reported are median values over 10 runs, as well as IQR in paranthesis.

**Distortions on the Instance 1 ($\epsilon = 0.05$)**

| $k$ | Coreset Size | Sensitivity Sampling | Uniform Sampling |
|-----|--------------|----------------------|------------------|
| 10  | $k^{1.00}/10\epsilon^2$ | 1.0010 (0.00010) | 1.0010 (0.00010) |
|     | $k^{1.25}/10\epsilon^2$ | 1.0009 (0.00065) | 1.0009 (0.00066) |
|     | $k^{1.50}/10\epsilon^2$ | 1.0005 (0.00066) | 1.0005 (0.00067) |
| 20  | $k^{1.00}/10\epsilon^2$ | 1.0007 (0.00027) | 1.0007 (0.00029) |
|     | $k^{1.25}/10\epsilon^2$ | 1.0004 (0.00048) | 1.0004 (0.00048) |
|     | $k^{1.50}/10\epsilon^2$ | 1.0003 (0.00031) | 1.0002 (0.00029) |
| 50  | $k^{1.00}/10\epsilon^2$ | 1.0003 (0.00034) | 1.0003 (0.00033) |
|     | $k^{1.25}/10\epsilon^2$ | 1.0002 (0.00020) | 1.0002 (0.00019) |
|     | $k^{1.50}/10\epsilon^2$ | 1.0001 (0.00012) | 1.0001 (0.00012) |
| 70  | $k^{1.00}/10\epsilon^2$ | 1.0002 (0.00026) | 1.0002 (0.00026) |
|     | $k^{1.25}/10\epsilon^2$ | 1.0001 (0.00015) | 1.0001 (0.00015) |
|     | $k^{1.50}/10\epsilon^2$ | 1.0001 (0.00009) | 1.0001 (0.00009) |
| 100 | $k^{1.00}/10\epsilon^2$ | 1.0002 (0.00021) | 1.0002 (0.00022) |
|     | $k^{1.25}/10\epsilon^2$ | 1.0001 (0.00011) | 1.0001 (0.00011) |
|     | $k^{1.50}/10\epsilon^2$ | 1.0001 (0.00006) | 1.0000 (0.00006) |

Table 6: Distortions of the evaluated algorithms on Instance 1 with $\epsilon = 0.05$. Reported are median values over 100 runs, as well as IQR in paranthesis.

**Distortions on the Instance 1 ($\epsilon = 0.1$)**

| $k$ | Coreset Size | Sensitivity Sampling | Uniform Sampling |
|-----|--------------|----------------------|------------------|
| 10  | $k^{1.00}/10\epsilon^2$ | 1.0038 (0.00040) | 1.0038 (0.00041) |
|     | $k^{1.25}/10\epsilon^2$ | 1.0036 (0.00249) | 1.0036 (0.00254) |
|     | $k^{1.50}/10\epsilon^2$ | 1.0021 (0.00267) | 1.0019 (0.00262) |
| 20  | $k^{1.00}/10\epsilon^2$ | 1.0026 (0.00123) | 1.0027 (0.00116) |
|     | $k^{1.25}/10\epsilon^2$ | 1.0015 (0.00192) | 1.0014 (0.00195) |
|     | $k^{1.50}/10\epsilon^2$ | 1.0010 (0.00123) | 1.0010 (0.00122) |
| 50  | $k^{1.00}/10\epsilon^2$ | 1.0013 (0.00130) | 1.0013 (0.00132) |
|     | $k^{1.25}/10\epsilon^2$ | 1.0007 (0.00080) | 1.0006 (0.00079) |
|     | $k^{1.50}/10\epsilon^2$ | 1.0004 (0.00050) | 1.0004 (0.00048) |
| 70  | $k^{1.00}/10\epsilon^2$ | 1.0009 (0.00106) | 1.0009 (0.00104) |
|     | $k^{1.25}/10\epsilon^2$ | 1.0005 (0.00059) | 1.0005 (0.00060) |
|     | $k^{1.50}/10\epsilon^2$ | 1.0003 (0.00035) | 1.0003 (0.00034) |
| 100 | $k^{1.00}/10\epsilon^2$ | 1.0006 (0.00086) | 1.0006 (0.00088) |
|     | $k^{1.25}/10\epsilon^2$ | 1.0004 (0.00045) | 1.0004 (0.00043) |
|     | $k^{1.50}/10\epsilon^2$ | 1.0002 (0.00024) | 1.0002 (0.00023) |

Table 7: Distortions of the evaluated algorithms on Instance 1 with $\epsilon = 0.1$. Reported are median values over 100 runs, as well as IQR in paranthesis.

**Distortions on the Instance 1 ($\epsilon = 0.2$)**

| $k$ | Coreset Size | Sensitivity Sampling | Uniform Sampling |
|-----|--------------|----------------------|-------------------|
| 10 | $k^{1.00}/10\epsilon^2$ | 1.0155 (0.00170) | 1.0155 (0.00173) |
| | $k^{1.25}/10\epsilon^2$ | 1.0147 (0.01015) | 1.0144 (0.01037) |
| | $k^{1.50}/10\epsilon^2$ | 1.0079 (0.01091) | 1.0077 (0.01067) |
| 20 | $k^{1.00}/10\epsilon^2$ | 1.0106 (0.00428) | 1.0106 (0.00422) |
| | $k^{1.25}/10\epsilon^2$ | 1.0060 (0.00770) | 1.0058 (0.00769) |
| | $k^{1.50}/10\epsilon^2$ | 1.0042 (0.00506) | 1.0039 (0.00473) |
| 50 | $k^{1.00}/10\epsilon^2$ | 1.0053 (0.00526) | 1.0052 (0.00527) |
| | $k^{1.25}/10\epsilon^2$ | 1.0027 (0.00315) | 1.0026 (0.00316) |
| | $k^{1.50}/10\epsilon^2$ | 1.0016 (0.00194) | 1.0015 (0.00189) |
| 70 | $k^{1.00}/10\epsilon^2$ | 1.0037 (0.00414) | 1.0035 (0.00422) |
| | $k^{1.25}/10\epsilon^2$ | 1.0020 (0.00251) | 1.0020 (0.00230) |
| | $k^{1.50}/10\epsilon^2$ | 1.0012 (0.00143) | 1.0011 (0.00139) |
| 100 | $k^{1.00}/10\epsilon^2$ | 1.0025 (0.00348) | 1.0025 (0.00349) |
| | $k^{1.25}/10\epsilon^2$ | 1.0015 (0.00183) | 1.0014 (0.00175) |
| | $k^{1.50}/10\epsilon^2$ | 1.0008 (0.00101) | 1.0008 (0.00094) |

Table 8: Distortions of the evaluated algorithms on Instance 1 with $\epsilon = 0.2$. Reported are median values over 100 runs, as well as IQR in paranthesis.

## C.2 Synthetic Instance 2

**Costs on the Instance 2 ($\epsilon = 0.01$)**

| $k$ | Coreset Size | Sensitivity Sampling | Uniform Sampling |
|-----|--------------|----------------------|-------------------|
| 10 | $k^{1.00}/10\epsilon^2$ | 7.532e+09 (2.516e+09) | 7.531e+09 (2.515e+09) |
| | $k^{1.25}/10\epsilon^2$ | 7.535e+09 (2.514e+09) | 7.532e+09 (2.515e+09) |
| | $k^{1.50}/10\epsilon^2$ | 7.533e+09 (2.516e+09) | 7.531e+09 (2.517e+09) |
| 20 | $k^{1.00}/10\epsilon^2$ | 2.489e+10 (3.562e+09) | 2.489e+10 (3.561e+09) |
| | $k^{1.25}/10\epsilon^2$ | 2.490e+10 (3.560e+09) | 2.137e+10 (3.559e+09) |
| | $k^{1.50}/10\epsilon^2$ | 2.488e+10 (3.562e+09) | 2.137e+10 (3.561e+09) |
| 50 | $k^{1.00}/10\epsilon^2$ | 1.070e+11 (5.644e+09) | 1.070e+11 (5.651e+09) |
| | $k^{1.25}/10\epsilon^2$ | 1.070e+11 (5.643e+09) | 1.070e+11 (5.641e+09) |
| | $k^{1.50}/10\epsilon^2$ | 1.070e+11 (5.637e+09) | 1.070e+11 (5.642e+09) |
| 70 | $k^{1.00}/10\epsilon^2$ | 1.867e+11 (6.677e+09) | 1.867e+11 (6.672e+09) |
| | $k^{1.25}/10\epsilon^2$ | 1.867e+11 (6.678e+09) | 1.867e+11 (6.675e+09) |
| | $k^{1.50}/10\epsilon^2$ | 1.867e+11 (6.677e+09) | 1.867e+11 (6.679e+09) |
| 100 | $k^{1.00}/10\epsilon^2$ | 3.270e+11 (7.990e+09) | 3.269e+11 (7.990e+09) |
| | $k^{1.25}/10\epsilon^2$ | 3.270e+11 (7.998e+09) | 3.270e+11 (8.015e+09) |
| | $k^{1.50}/10\epsilon^2$ | 3.269e+11 (7.992e+09) | 3.270e+11 (7.987e+09) |

Table 9: Costs of the evaluated algorithms on Instance 2 with $\epsilon = 0.01$. Reported are median values over 10 runs, as well as IQR in paranthesis.

**Costs on the Instance 2 ($\epsilon = 0.05$)**

| $k$ | Coreset Size | Sensitivity Sampling | Uniform Sampling |
|-----|--------------|----------------------|-------------------|
| 10 | $k^{1.00}/10\epsilon^2$ | 1.172e+07 (3.931e+06) | 1.171e+07 (3.923e+06) |
| | $k^{1.25}/10\epsilon^2$ | 1.171e+07 (3.929e+06) | 1.172e+07 (3.930e+06) |
| | $k^{1.50}/10\epsilon^2$ | 1.171e+07 (3.930e+06) | 1.172e+07 (3.928e+06) |
| 20 | $k^{1.00}/10\epsilon^2$ | 3.893e+07 (5.572e+06) | 3.379e+07 (5.570e+06) |
| | $k^{1.25}/10\epsilon^2$ | 3.890e+07 (5.569e+06) | 3.380e+07 (5.574e+06) |

| | $k^{1.50}/10\epsilon^2$ | 3.379e+07 (5.574e+06) | 3.894e+07 (5.577e+06) |
|---|---|---|---|
| 50 | $k^{1.00}/10\epsilon^2$ | 1.692e+08 (8.906e+06) | 1.692e+08 (8.895e+06) |
| | $k^{1.25}/10\epsilon^2$ | 1.691e+08 (8.889e+06) | 1.692e+08 (8.889e+06) |
| | $k^{1.50}/10\epsilon^2$ | 1.692e+08 (8.896e+06) | 1.692e+08 (8.919e+06) |
| 70 | $k^{1.00}/10\epsilon^2$ | 2.958e+08 (1.055e+07) | 2.957e+08 (1.053e+07) |
| | $k^{1.25}/10\epsilon^2$ | 2.957e+08 (1.056e+07) | 2.957e+08 (1.056e+07) |
| | $k^{1.50}/10\epsilon^2$ | 2.958e+08 (1.054e+07) | 2.958e+08 (1.056e+07) |
| 100 | $k^{1.00}/10\epsilon^2$ | 5.192e+08 (1.268e+07) | 5.192e+08 (1.268e+07) |
| | $k^{1.25}/10\epsilon^2$ | 5.192e+08 (1.267e+07) | 5.192e+08 (1.268e+07) |
| | $k^{1.50}/10\epsilon^2$ | 5.192e+08 (1.267e+07) | 5.192e+08 (1.269e+07) |

Table 10: Costs of the evaluated algorithms on Instance 2 with $\epsilon = 0.05$. Reported are median values over 100 runs, as well as IQR in paranthesis.

**Costs on the Instance 2 ($\epsilon = 0.1$)**

| $k$ | Coreset Size | Sensitivity Sampling | Uniform Sampling |
|---|---|---|---|
| 10 | $k^{1.00}/10\epsilon^2$ | 7.077e+05 (2.361e+05) | 7.058e+05 (2.360e+05) |
| | $k^{1.25}/10\epsilon^2$ | 7.090e+05 (2.367e+05) | 7.080e+05 (2.363e+05) |
| | $k^{1.50}/10\epsilon^2$ | 7.058e+05 (2.369e+05) | 7.070e+05 (2.359e+05) |
| 20 | $k^{1.00}/10\epsilon^2$ | 2.097e+06 (3.381e+05) | 2.097e+06 (3.377e+05) |
| | $k^{1.25}/10\epsilon^2$ | 2.097e+06 (3.371e+05) | 2.106e+06 (3.375e+05) |
| | $k^{1.50}/10\epsilon^2$ | 2.099e+06 (3.374e+05) | 2.374e+06 (3.367e+05) |
| 50 | $k^{1.00}/10\epsilon^2$ | 1.052e+07 (5.438e+05) | 1.051e+07 (5.424e+05) |
| | $k^{1.25}/10\epsilon^2$ | 1.052e+07 (5.443e+05) | 1.051e+07 (5.425e+05) |
| | $k^{1.50}/10\epsilon^2$ | 1.051e+07 (5.435e+05) | 1.052e+07 (5.452e+05) |
| 70 | $k^{1.00}/10\epsilon^2$ | 1.846e+07 (6.438e+05) | 1.846e+07 (6.450e+05) |
| | $k^{1.25}/10\epsilon^2$ | 1.846e+07 (6.456e+05) | 1.846e+07 (6.443e+05) |
| | $k^{1.50}/10\epsilon^2$ | 1.846e+07 (6.437e+05) | 1.846e+07 (6.441e+05) |
| 100 | $k^{1.00}/10\epsilon^2$ | 3.255e+07 (7.797e+05) | 3.255e+07 (7.771e+05) |
| | $k^{1.25}/10\epsilon^2$ | 3.255e+07 (7.779e+05) | 3.255e+07 (7.793e+05) |
| | $k^{1.50}/10\epsilon^2$ | 3.255e+07 (7.760e+05) | 3.255e+07 (7.786e+05) |

Table 11: Costs of the evaluated algorithms on Instance 2 with $\epsilon = 0.1$. Reported are median values over 100 runs, as well as IQR in paranthesis.

**Costs on the Instance 2 ($\epsilon = 0.2$)**

| $k$ | Coreset Size | Sensitivity Sampling | Uniform Sampling |
|---|---|---|---|
| 10 | $k^{1.00}/10\epsilon^2$ | 4.257e+04 (1.354e+04) | 4.239e+04 (1.362e+04) |
| | $k^{1.25}/10\epsilon^2$ | 4.266e+04 (1.361e+04) | 4.270e+04 (1.355e+04) |
| | $k^{1.50}/10\epsilon^2$ | 4.247e+04 (1.365e+04) | 4.240e+04 (1.354e+04) |
| 20 | $k^{1.00}/10\epsilon^2$ | 1.340e+05 (1.922e+04) | 1.347e+05 (1.937e+04) |
| | $k^{1.25}/10\epsilon^2$ | 1.338e+05 (1.935e+04) | 1.343e+05 (1.937e+04) |
| | $k^{1.50}/10\epsilon^2$ | 1.464e+05 (1.933e+04) | 1.465e+05 (1.949e+04) |
| 50 | $k^{1.00}/10\epsilon^2$ | 6.702e+05 (3.148e+04) | 6.704e+05 (3.152e+04) |
| | $k^{1.25}/10\epsilon^2$ | 6.703e+05 (3.161e+04) | 6.703e+05 (3.138e+04) |
| | $k^{1.50}/10\epsilon^2$ | 6.703e+05 (3.143e+04) | 6.705e+05 (3.155e+04) |
| 70 | $k^{1.00}/10\epsilon^2$ | 1.190e+06 (3.747e+04) | 1.191e+06 (3.738e+04) |
| | $k^{1.25}/10\epsilon^2$ | 1.190e+06 (3.749e+04) | 1.190e+06 (3.735e+04) |

|     |                        |                      |                      |
| --- | ---------------------- | -------------------- | -------------------- |
|     | $k^{1.50}/10\epsilon^2$ | 1.190e+06 (3.721e+04) | 1.190e+06 (3.740e+04) |
| 100 | $k^{1.00}/10\epsilon^2$ | 2.127e+06 (4.514e+04) | 2.126e+06 (4.517e+04) |
|     | $k^{1.25}/10\epsilon^2$ | 2.127e+06 (4.527e+04) | 2.127e+06 (4.583e+04) |
|     | $k^{1.50}/10\epsilon^2$ | 2.127e+06 (4.526e+04) | 2.127e+06 (4.553e+04) |

Table 12: Costs of the evaluated algorithms on Instance 2 with $\epsilon = 0.2$. Reported are median values over 100 runs, as well as IQR in paranthesis.

**Distortions on the Instance 2 ($\epsilon = 0.01$)**

| $k$ | Coreset Size | Sensitivity Sampling | Uniform Sampling |
| --- | --- | --- | --- |
| 10 | $k^{1.00}/10\epsilon^2$ | 1.0084 (0.01206) | 1.0076 (0.00832) |
|    | $k^{1.25}/10\epsilon^2$ | 1.0060 (0.00749) | 1.0053 (0.00603) |
|    | $k^{1.50}/10\epsilon^2$ | 1.0048 (0.00562) | 1.0038 (0.00453) |
| 20 | $k^{1.00}/10\epsilon^2$ | 1.0033 (0.00439) | 1.0032 (0.00407) |
|    | $k^{1.25}/10\epsilon^2$ | 1.0027 (0.00317) | 1.0029 (0.00386) |
|    | $k^{1.50}/10\epsilon^2$ | 1.0018 (0.00263) | 1.0016 (0.00198) |
| 50 | $k^{1.00}/10\epsilon^2$ | 1.0017 (0.00218) | 1.0019 (0.00224) |
|    | $k^{1.25}/10\epsilon^2$ | 1.0011 (0.00127) | 1.0012 (0.00141) |
|    | $k^{1.50}/10\epsilon^2$ | 1.0006 (0.00084) | 1.0006 (0.00074) |
| 70 | $k^{1.00}/10\epsilon^2$ | 1.0013 (0.00164) | 1.0012 (0.00156) |
|    | $k^{1.25}/10\epsilon^2$ | 1.0007 (0.00091) | 1.0007 (0.00087) |
|    | $k^{1.50}/10\epsilon^2$ | 1.0005 (0.00060) | 1.0005 (0.00051) |
| 100 | $k^{1.00}/10\epsilon^2$ | 1.0010 (0.00142) | 1.0009 (0.00124) |
|    | $k^{1.25}/10\epsilon^2$ | 1.0006 (0.00074) | 1.0005 (0.00069) |
|    | $k^{1.50}/10\epsilon^2$ | 1.0003 (0.00039) | 1.0003 (0.00035) |

Table 13: Distortions of the evaluated algorithms on Instance 2 with $\epsilon = 0.01$. Reported are median values over 10 runs, as well as IQR in paranthesis.

**Distortions on the Instance 2 ($\epsilon = 0.05$)**

| $k$ | Coreset Size | Sensitivity Sampling | Uniform Sampling |
| --- | --- | --- | --- |
| 10 | $k^{1.00}/10\epsilon^2$ | 1.0394 (0.05072) | 1.0330 (0.04356) |
|    | $k^{1.25}/10\epsilon^2$ | 1.0302 (0.04054) | 1.0228 (0.03016) |
|    | $k^{1.50}/10\epsilon^2$ | 1.0238 (0.03132) | 1.0172 (0.02158) |
| 20 | $k^{1.00}/10\epsilon^2$ | 1.0179 (0.02300) | 1.0177 (0.02212) |
|    | $k^{1.25}/10\epsilon^2$ | 1.0130 (0.01635) | 1.0122 (0.01491) |
|    | $k^{1.50}/10\epsilon^2$ | 1.0087 (0.01071) | 1.0076 (0.00974) |
| 50 | $k^{1.00}/10\epsilon^2$ | 1.0089 (0.01142) | 1.0085 (0.01062) |
|    | $k^{1.25}/10\epsilon^2$ | 1.0053 (0.00661) | 1.0050 (0.00624) |
|    | $k^{1.50}/10\epsilon^2$ | 1.0033 (0.00409) | 1.0031 (0.00401) |
| 70 | $k^{1.00}/10\epsilon^2$ | 1.0064 (0.00813) | 1.0064 (0.00820) |
|    | $k^{1.25}/10\epsilon^2$ | 1.0038 (0.00480) | 1.0038 (0.00463) |
|    | $k^{1.50}/10\epsilon^2$ | 1.0021 (0.00271) | 1.0021 (0.00262) |
| 100 | $k^{1.00}/10\epsilon^2$ | 1.0051 (0.00620) | 1.0049 (0.00610) |
|    | $k^{1.25}/10\epsilon^2$ | 1.0028 (0.00354) | 1.0027 (0.00336) |
|    | $k^{1.50}/10\epsilon^2$ | 1.0016 (0.00205) | 1.0015 (0.00186) |

Table 14: Distortions of the evaluated algorithms on Instance 2 with $\epsilon = 0.05$. Reported are median values over 100 runs, as well as IQR in paranthesis.

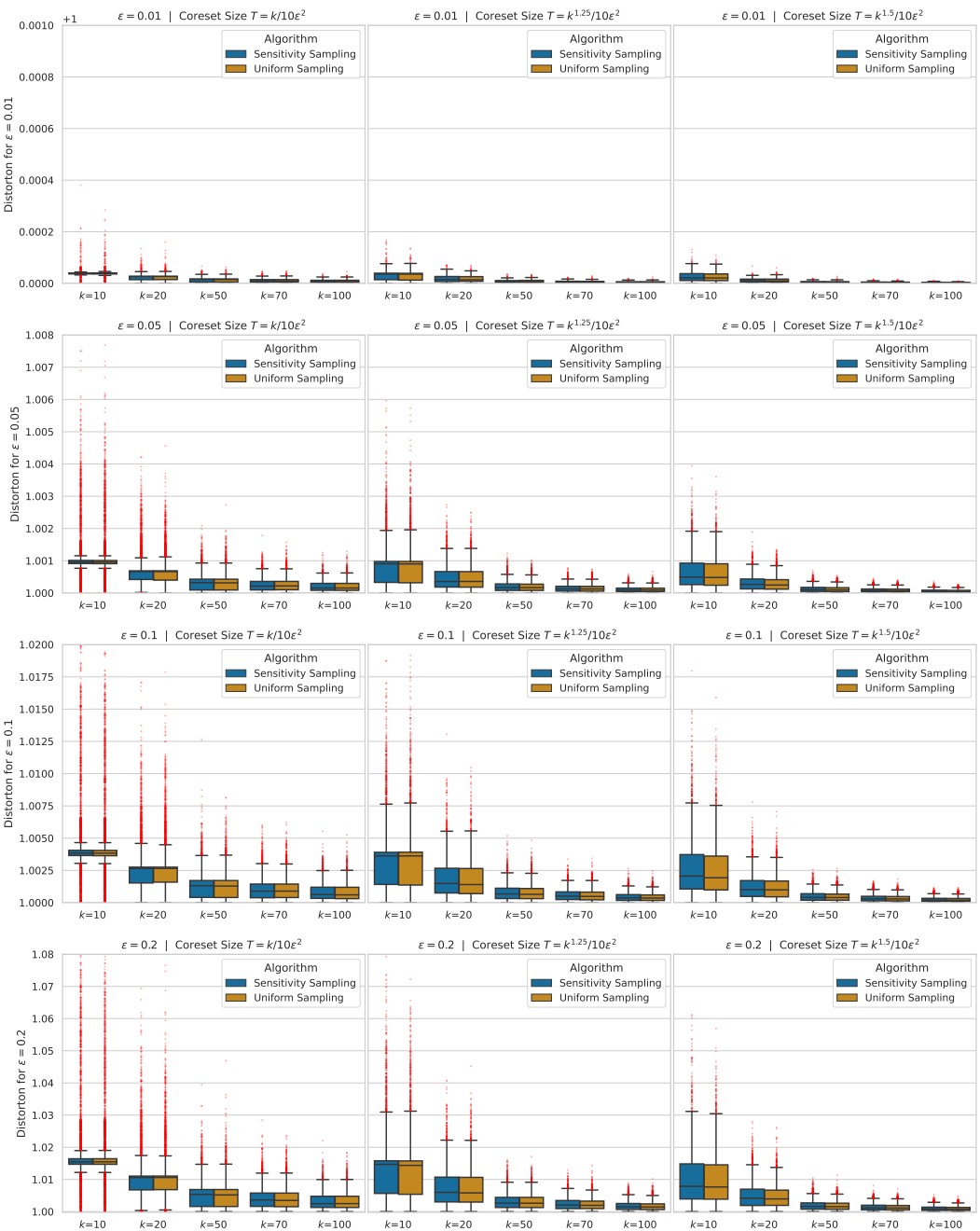

Figure 2: Distortions of sensitivity sampling and uniform sampling on instance 1 for different values of $\epsilon$. The red dots represent outlier observations, i.e. values outside 1.5 times IQR below the 25th percentile and above the 75th percentile.

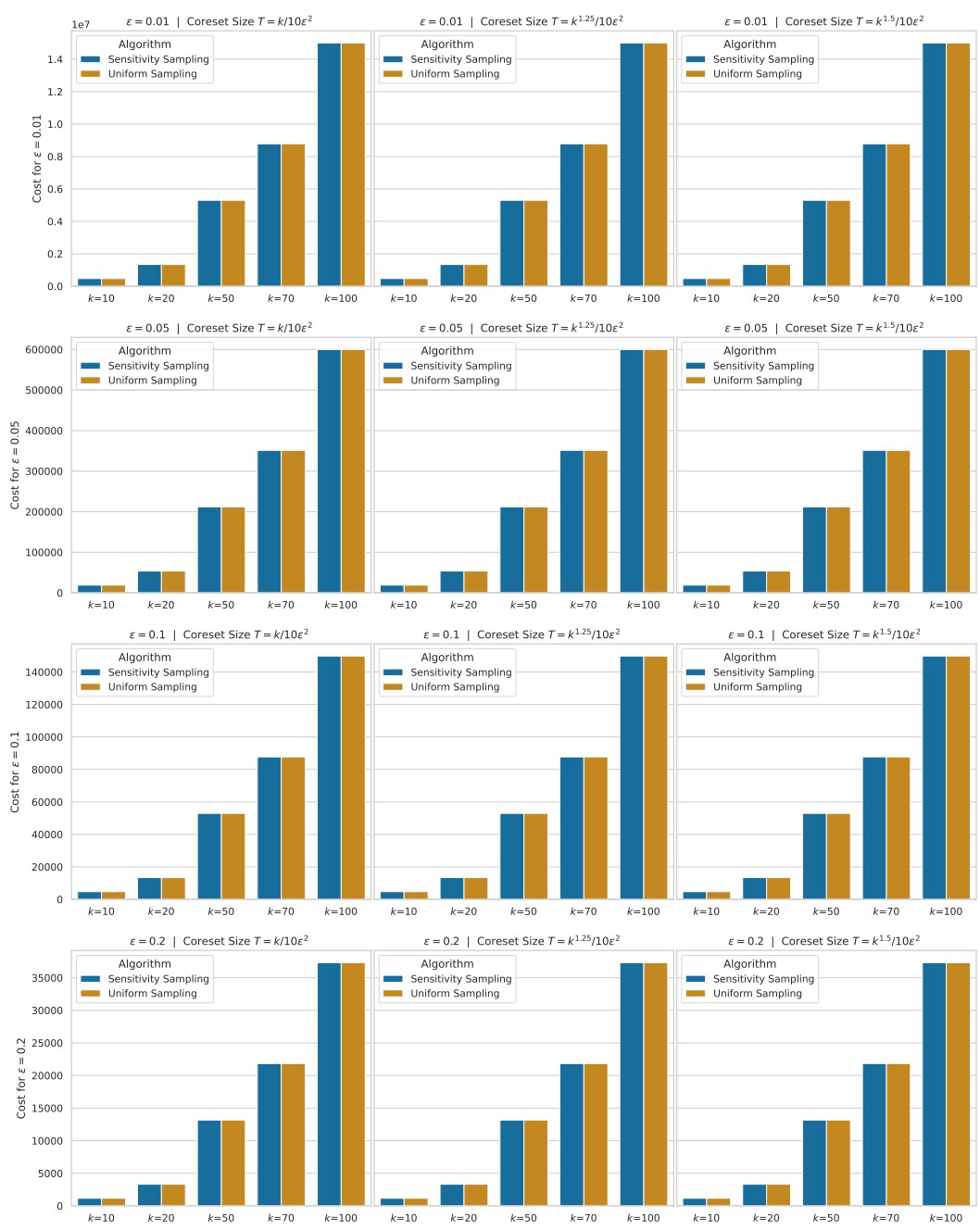

Figure 3: Costs of sensitivity sampling and uniform sampling on instance 1 for different values of $\epsilon$.

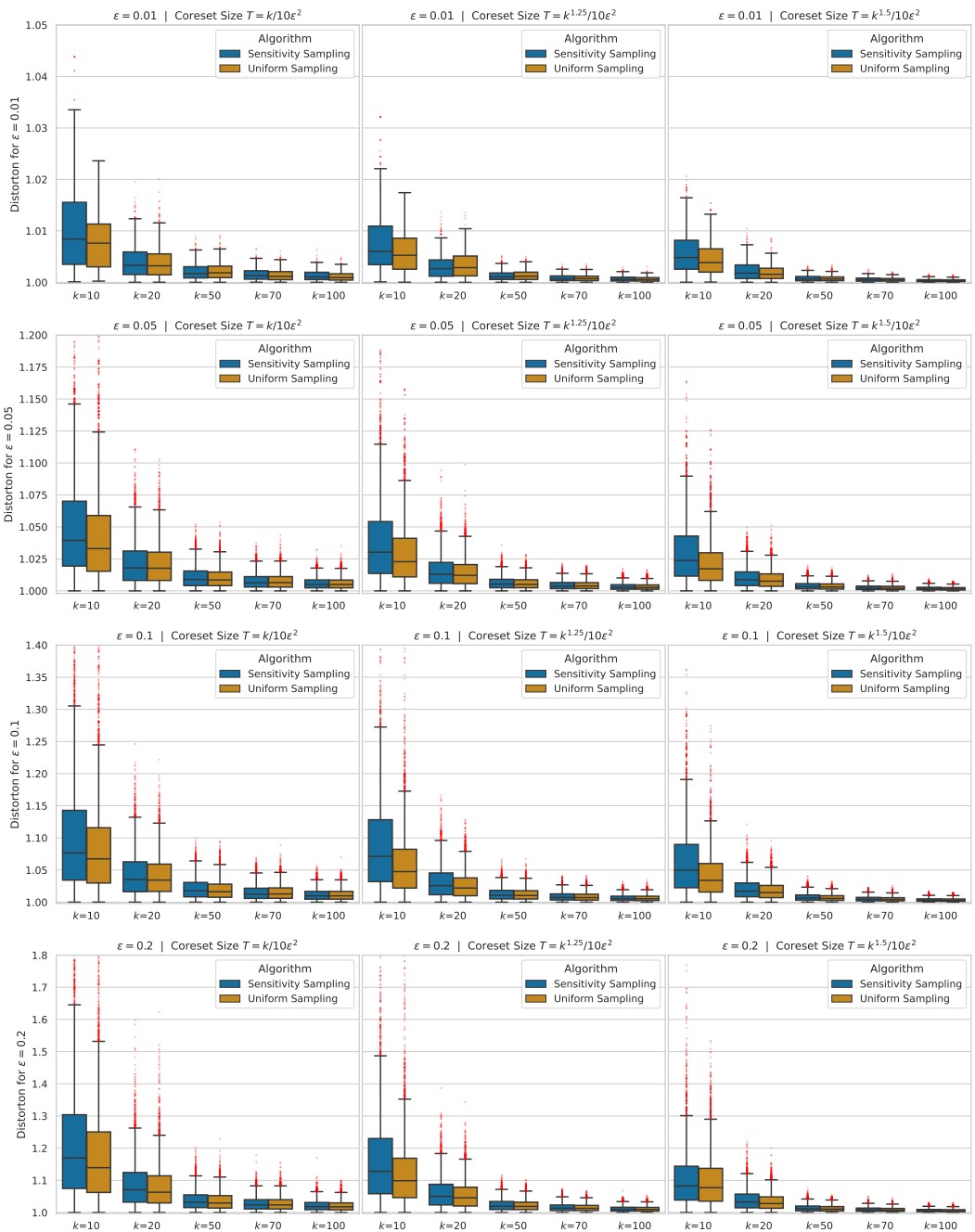

Figure 4: Distortions of sensitivity sampling and uniform sampling on instance 2 for different values of $\epsilon$. The red dots represent outlier observations, i.e. values outside 1.5 times IQR below the 25th percentile and above the 75th percentile.

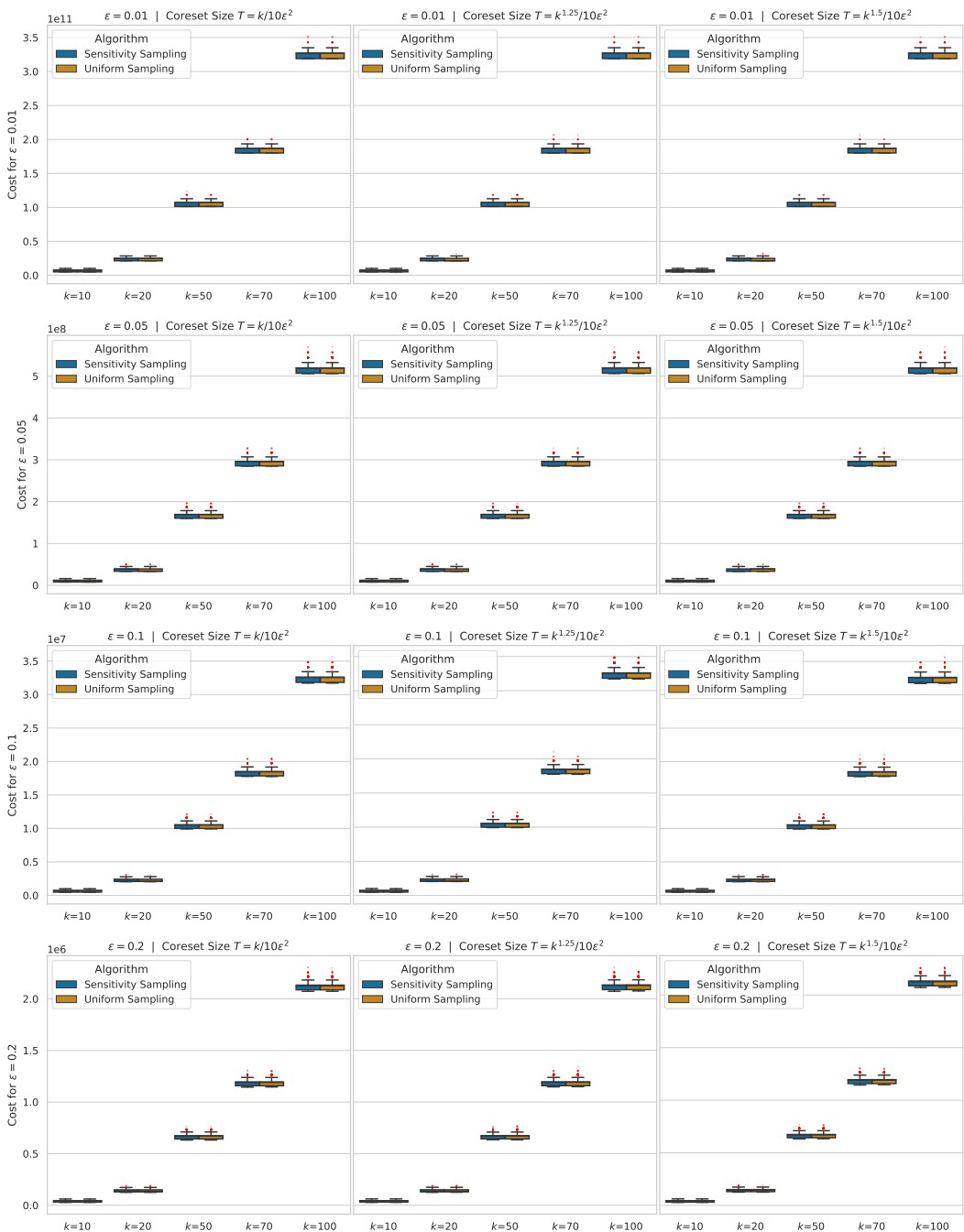

Figure 5: Costs of sensitivity sampling and uniform sampling on instance 2 for different values of $\epsilon$. The red dots represent outlier observations, i.e. values outside 1.5 times IQR below the 25th percentile and above the 75th percentile.

**Distortions on the Instance 2 ($\epsilon = 0.1$)**

| $k$ | Coreset Size | Sensitivity Sampling | Uniform Sampling |
|---|---|---|---|
| 10 | $k^{1.00}/10\epsilon^2$ | 1.0765 (0.10850) | 1.0674 (0.08599) |
|  | $k^{1.25}/10\epsilon^2$ | 1.0713 (0.09632) | 1.0476 (0.06037) |
|  | $k^{1.50}/10\epsilon^2$ | 1.0499 (0.06741) | 1.0339 (0.04438) |
| 20 | $k^{1.00}/10\epsilon^2$ | 1.0353 (0.04650) | 1.0343 (0.04253) |
|  | $k^{1.25}/10\epsilon^2$ | 1.0258 (0.03387) | 1.0220 (0.02776) |
|  | $k^{1.50}/10\epsilon^2$ | 1.0173 (0.02142) | 1.0149 (0.01897) |
| 50 | $k^{1.00}/10\epsilon^2$ | 1.0177 (0.02241) | 1.0162 (0.02037) |
|  | $k^{1.25}/10\epsilon^2$ | 1.0106 (0.01339) | 1.0104 (0.01295) |
|  | $k^{1.50}/10\epsilon^2$ | 1.0066 (0.00814) | 1.0058 (0.00736) |
| 70 | $k^{1.00}/10\epsilon^2$ | 1.0124 (0.01584) | 1.0128 (0.01617) |
|  | $k^{1.25}/10\epsilon^2$ | 1.0073 (0.00944) | 1.0071 (0.00917) |
|  | $k^{1.50}/10\epsilon^2$ | 1.0044 (0.00542) | 1.0041 (0.00497) |
| 100 | $k^{1.00}/10\epsilon^2$ | 1.0098 (0.01209) | 1.0098 (0.01202) |
|  | $k^{1.25}/10\epsilon^2$ | 1.0055 (0.00670) | 1.0054 (0.00676) |
|  | $k^{1.50}/10\epsilon^2$ | 1.0030 (0.00369) | 1.0029 (0.00358) |

Table 15: Distortions of the evaluated algorithms on Instance 2 with $\epsilon = 0.1$. Reported are median values over 100 runs, as well as IQR in paranthesis.

**Distortions on the Instance 2 ($\epsilon = 0.2$)**

| $k$ | Coreset Size | Sensitivity Sampling | Uniform Sampling |
|---|---|---|---|
| 10 | $k^{1.00}/10\epsilon^2$ | 1.1699 (0.22919) | 1.1394 (0.18815) |
|  | $k^{1.25}/10\epsilon^2$ | 1.1280 (0.17168) | 1.0988 (0.12250) |
|  | $k^{1.50}/10\epsilon^2$ | 1.0827 (0.10521) | 1.0772 (0.10201) |
| 20 | $k^{1.00}/10\epsilon^2$ | 1.0710 (0.09212) | 1.0631 (0.08432) |
|  | $k^{1.25}/10\epsilon^2$ | 1.0502 (0.06418) | 1.0457 (0.05854) |
|  | $k^{1.50}/10\epsilon^2$ | 1.0328 (0.04265) | 1.0278 (0.03545) |
| 50 | $k^{1.00}/10\epsilon^2$ | 1.0320 (0.03946) | 1.0296 (0.03859) |
|  | $k^{1.25}/10\epsilon^2$ | 1.0197 (0.02511) | 1.0190 (0.02300) |
|  | $k^{1.50}/10\epsilon^2$ | 1.0116 (0.01466) | 1.0106 (0.01364) |
| 70 | $k^{1.00}/10\epsilon^2$ | 1.0237 (0.02871) | 1.0236 (0.02887) |
|  | $k^{1.25}/10\epsilon^2$ | 1.0136 (0.01707) | 1.0129 (0.01588) |
|  | $k^{1.50}/10\epsilon^2$ | 1.0080 (0.00992) | 1.0077 (0.00906) |
| 100 | $k^{1.00}/10\epsilon^2$ | 1.0184 (0.02253) | 1.0177 (0.02165) |
|  | $k^{1.25}/10\epsilon^2$ | 1.0095 (0.01185) | 1.0094 (0.01151) |
|  | $k^{1.50}/10\epsilon^2$ | 1.0056 (0.00674) | 1.0053 (0.00638) |

Table 16: Distortions of the evaluated algorithms on Instance 2 with $\epsilon = 0.2$. Reported are median values over 100 runs, as well as IQR in paranthesis.

## C.3 Real-world Data Sets

**Costs on the Census**

| $k$ | Group Sampling | Sensitivity Sampling | Uniform Sampling |
|---|---|---|---|
| 10 | 2.54e+08 (7.73e+06) | 2.57e+08 (1.20e+07) | 2.50e+08 (4.27e+06) |
| 20 | 1.90e+08 (5.27e+06) | 1.93e+08 (3.73e+06) | 1.91e+08 (2.83e+06) |
| 30 | 1.63e+08 (3.99e+06) | 1.64e+08 (3.96e+06) | 1.60e+08 (2.69e+06) |
| 40 | 1.42e+08 (2.15e+06) | 1.44e+08 (5.00e+06) | 1.42e+08 (1.51e+06) |

| 50 | 1.28e+08 (1.75e+06) | 1.32e+08 (3.72e+06) | 1.29e+08 (1.41e+06) |

Table 17: Costs of the evaluated algorithms on the Census.

**Costs on the Covertype**

| $k$ | Group Sampling | Sensitivity Sampling | Uniform Sampling |
|---|---|---|---|
| 10 | 3.45e+11 (2.53e+09) | 3.44e+11 (2.79e+09) | 3.47e+11 (3.56e+09) |
| 20 | 2.07e+11 (2.31e+09) | 2.07e+11 (1.93e+09) | 2.06e+11 (1.08e+09) |
| 30 | 1.57e+11 (1.34e+09) | 1.57e+11 (8.25e+08) | 1.57e+11 (1.22e+09) |
| 40 | 1.31e+11 (6.80e+08) | 1.31e+11 (6.02e+08) | 1.31e+11 (3.10e+08) |
| 50 | 1.15e+11 (1.10e+09) | 1.15e+11 (8.82e+08) | 1.15e+11 (6.26e+08) |

Table 18: Costs of the evaluated algorithms on the Covertype.

**Costs on the Tower**

| $k$ | Group Sampling | Sensitivity Sampling | Uniform Sampling |
|---|---|---|---|
| 20 | 6.33e+08 (1.34e+07) | 6.42e+08 (1.98e+07) | 6.41e+08 (1.49e+07) |
| 40 | 3.38e+08 (8.29e+06) | 3.40e+08 (7.63e+06) | 3.40e+08 (2.61e+06) |
| 60 | 2.44e+08 (1.63e+06) | 2.46e+08 (2.57e+06) | 2.48e+08 (1.56e+06) |
| 80 | 1.96e+08 (2.54e+06) | 1.98e+08 (1.91e+06) | 1.99e+08 (2.14e+06) |
| 100 | 1.66e+08 (8.34e+05) | 1.67e+08 (1.39e+06) | 1.68e+08 (1.12e+06) |

Table 19: Costs of the evaluated algorithms on the Tower.

**Distortions on the Census**

| $k$ | Group Sampling | Sensitivity Sampling | Uniform Sampling |
|---|---|---|---|
| 10 | 1.02e+00 (8.56e-03) | 1.01e+00 (6.83e-03) | 1.03e+00 (2.16e-02) |
| 20 | 1.02e+00 (9.85e-03) | 1.01e+00 (5.55e-03) | 1.02e+00 (1.40e-02) |
| 30 | 1.02e+00 (5.66e-03) | 1.00e+00 (4.14e-03) | 1.02e+00 (1.12e-02) |
| 40 | 1.02e+00 (3.56e-03) | 1.00e+00 (2.87e-03) | 1.02e+00 (9.32e-03) |
| 50 | 1.03e+00 (7.10e-03) | 1.01e+00 (1.68e-03) | 1.02e+00 (1.16e-02) |

Table 20: Distortions of the evaluated algorithms on the Census.

**Distortions on the Covertype**

| $k$ | Group Sampling | Sensitivity Sampling | Uniform Sampling |
|---|---|---|---|
| 10 | 1.03e+00 (1.50e-02) | 1.01e+00 (9.04e-03) | 1.04e+00 (1.98e-02) |
| 20 | 1.02e+00 (8.78e-03) | 1.01e+00 (7.80e-03) | 1.03e+00 (9.43e-03) |
| 30 | 1.02e+00 (4.94e-03) | 1.01e+00 (6.16e-03) | 1.04e+00 (8.46e-03) |
| 40 | 1.02e+00 (7.93e-03) | 1.00e+00 (4.95e-03) | 1.03e+00 (8.00e-03) |
| 50 | 1.02e+00 (5.39e-03) | 1.01e+00 (3.94e-03) | 1.04e+00 (7.82e-03) |

Table 21: Distortions of the evaluated algorithms on the Covertype.

**Distortions on the Tower**

| $k$ | Group Sampling | Sensitivity Sampling | Uniform Sampling |
|---|---|---|---|
| 20 | 1.02e+00 (1.13e-02) | 1.01e+00 (8.81e-03) | 1.04e+00 (3.78e-02) |
| 40 | 1.03e+00 (8.08e-03) | 1.01e+00 (6.54e-03) | 1.06e+00 (3.03e-02) |
| 60 | 1.03e+00 (5.37e-03) | 1.01e+00 (3.88e-03) | 1.06e+00 (1.74e-02) |
| 80 | 1.03e+00 (7.62e-03) | 1.00e+00 (1.86e-03) | 1.06e+00 (1.38e-02) |

| 100 | 1.03e+00 (5.89e-03) | 1.01e+00 (6.57e-03) | 1.06e+00 (2.20e-02) |

Table 22: Distortions of the evaluated algorithms on the Tower.

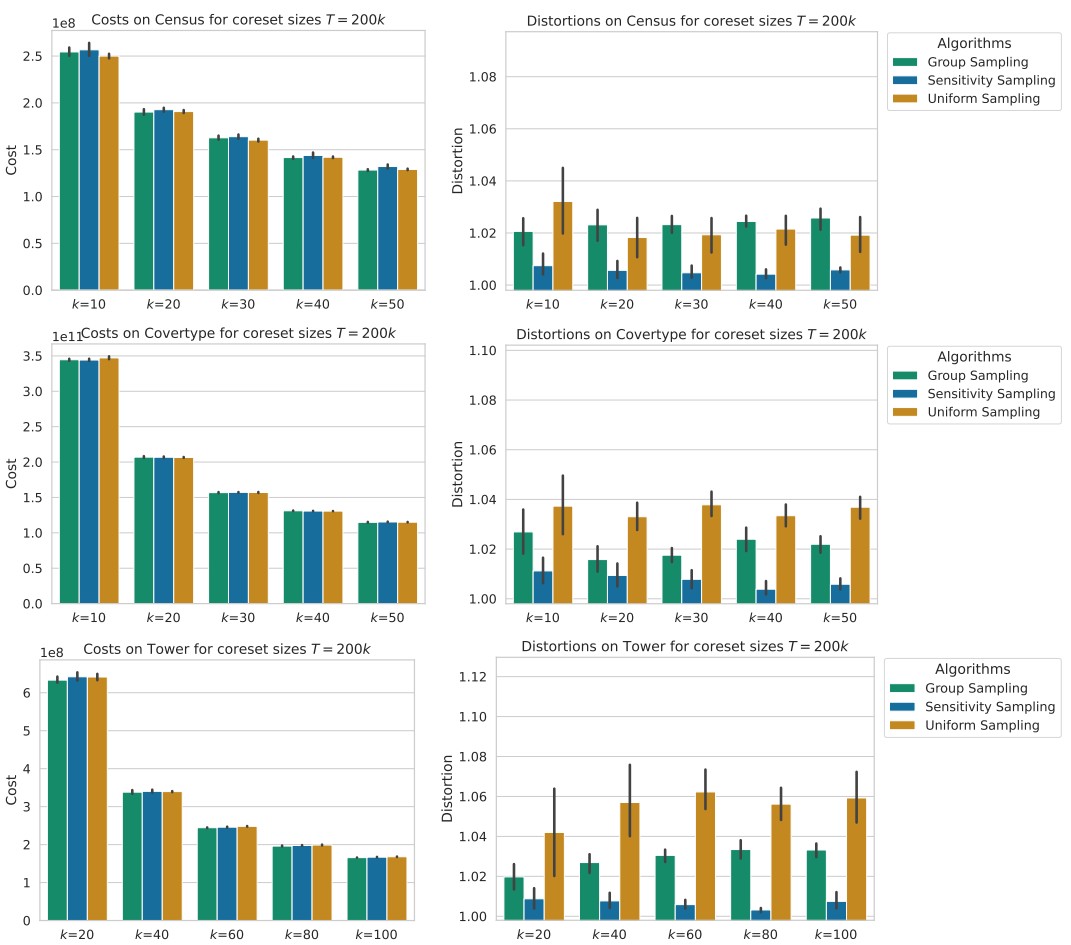

Figure 6: Costs and distortions on real-world data sets.