# OpenReview forum: "Improved Coresets for Euclidean $k$-Means"
_NeurIPS.cc/2022/Conference — NeurIPS 2022 Accept_

### Official Review · Reviewer_1Yz9 · 2022-07-07

**Rating:** 5
**Confidence:** 3
**Soundness:** 3 good
**Presentation:** 3 good
**Contribution:** 2 fair

**Summary:**

The authors address the problem of coreset construction for the known k-means problem. They propose a coreset construction scheme and prove that it achieves coresets of size smaller than the state of the art, in some regimes.
To carry out the analysis, the authors propose a novel method that avoids terminal embeddings, which are common in the analysis of prior works. The authors claim this novel technical contribution may be of independent interest as well.
Empirical evaluations are then carried out to test the impact of the proposed approach against known techniques.


**Questions:**

See few question above.

**Limitations:**

Yes

**Strengths And Weaknesses:**

Strengths:
- The proposed scheme indeed achieves state of the art coreset sizes for one of the cornerstone problems in the field.
- The analysis are novel and very interesting.
- The paper is relatively well written.

Weaknesses:
- Related work section is rushed through and lacks details. It does not contribute to the reader's knowledge, and should be better written in my opinion.
- Missing important discussions. Examples:
  * The validity of Assumptions 1-4 is placed in the appendix. At least a brief explanation should be placed in the main text.
  * A discussion as to why this result cannot be extended to l_z clustering can be very helpful and enriching.
  * It seems that the proposed algorithm, after ensuring the validity of the assumptions, samples each point according to some simple quantity which resembles the sensitivity distribution known from many prior works. A comparison or discussion is needed here.
- As of Assumption 3 regarding the input weights: to ensure integer weights (as stated in the appendix), some scaling and rounding is required. Doesnt this produce a dependency on the smallest weight, or, alternatively, on the model's precision? If so, this is an important note.
- The experimental results are not sufficient:
  * Important details are missing.
  * The proposed method is not shown to achieve better results than the competing methods.
  * Running time are not reported, which is crucial, as it seems that the proposed method would (theoretically) achieve much larger computational times. This is due to the computations required to enforce Assumptions 1-4 (which include a computation of an alternative coreset, e.g., the sensitivity coreset used as a competing method).
- At the end of the day, the proposed algorithm is not novel as it simply computes a (previously known) sampling probability for every input point, which is based on some (previously known) approximation for the problem, then it samples points from this distribution and does some reweighting.  Also, as the authors claim, enforcing Assumptions 1-4 is also not novel. The only novel part seems to be the analysis. Hence, in my opinion, the paper's main contribution is not sufficient.

Minor comments:
- The first paragraph on Page 2 is important and should be somehow better emphasized.
- Line 50: "may be reduced to log(k/\eps)\eps^-2 using dimension reduction techniques". some citation should be added at the end of this claim.
- Line 50: "to technique" (typo).
- Equation at Line 50: missing closing bracket.
- Typos at Line 63.
- The equation after Eq. (2) utilizes the point on the left and on the right hand sides, but p does not seem to be defined.
- Lemma 5: where is X defined?
- Line 210: the maximum distortion is reported in the experiments. Do you have insights on the average distortion? The average distortion might be a more accurate indicator.

---

> ### Author Response · Authors · 2022-07-28
> **Response to Reviewer 4**
>
> We wished to give strong evidence that the Group Sampling algorithm will lead to an optimal coreset bound, at least asymptotically up to $polylog(k/\varepsilon)$ factors. Finding an optimal coreset algorithm is inarguably the most important open question in coreset research. Group Sampling is the best candidate algorithm, giving that it currently has the best theoretical bounds, and the only contribution of a paper making progress on the optimality of Group Sampling is the analysis. The reviewer may consider the improvement in the bound as not being a result that is strong enough. Beating the $k^2$ bound was not mentioned by the reviewer once and it is not clear if the reviewer understands the significance of it. If so, we might have made that point clearer. But all the other reasonings (Group Sampling not being a new algorithm, giving more explainations how the preprocessing in Group Sampling works, explaining how Group Sampling compares to other sampling distributions) are fallacious. It is not possible to give a reader unfamiliar with coresets an introduction. For an introduction, there exist dedicated surveys which we will happily add to the paper.
>
> Details missing:
> For the specific complaint of group Sampling not being described: Given that Group Sampling was also used in two previous works, we decided to focus on the more novel aspects. While it might be preferable that all aspects of Group Sampling were given in this paper, the original paper by CSS'21 was over 50 pages, with roughly 20 pages describing how to canonize coreset constructions. Reviewing these assumptions in the main body is not possible if we are also to include our novel contributions, due to the page limit.
>
> Assumption 3 does not introduce any issues with bit complexity. Enforcing this assumption merely takes place in the analysis. In other words, an analysis for unweighted points implies an analysis with weighted points. The reference showing this was given in the supplementary material.
>
> $\ell_z$ clustering: see comment below.
>
> Comparing the estimator to previous sampling distributions is not reasonable, especially given that none of the distributions are new, as the reviewer states. Experts can assess the relationship between these distributions. In no paper has such a comparison ever taken place. Reviewing all sampling distributions would qualify as a detailed technical survey paper itself. We cannot do this here, especially given the page limit. If the reviewer has specific questions, we will answer them.
>
> Minor Comments:
> The $p$ after Equation 2 should be in $C_i$.
> $X$ in Lemma 5 should be the input point set $P$.

---

> ### Author Response · Authors · 2022-07-28
> **Experiments**
>
> As for the supposed poor experimental performance of Group Sampling vs Sensitivity Sampling: We were open about the fact that we believe that a good analysis of Sensitivity Sampling will also yield an optimal coreset or at least a bound that matches Group Sampling. In fact, given that Sensitivity Sampling is arguably easier, we and presumably other coreset researchers would welcome this. Unfortunately this is not known, despite the algorithm being older and arguably better studied. Moreover, the main point of the experiments was not to compare the two algorithms to each other, to analyse a data set, or to evaluate secondary characteristics of these algorithms, but *only* to study whether the coreset size should depend linearly on k or not. In our view, it is not conductive towards obtaining optimal coreset bounds if we refuse to study Group Sampling simply because Sensitivity Sampling is better empirically, especially given that Group Sampling has the better theoretical bounds. For both algorithms, the experiments showed that a linear dependecy on k seems to be the right answer.
>
> It is known that Group Sampling can be implemented efficiently, see for example the recent study https://arxiv.org/abs/2207.00966, though of course Sensitivity Sampling will always be slightly faster. Beyond that, running such experiments here does not add towards the important open question of obtaining optimal sized coresets. Running times, as long as they are nearly linear time (which Group Sampling supports) are not important for this paper.
>
> The question of average distortion does not make sense. Coresets are a worst case guarantee. If even a single solution has high distortion, then that is the distortion of the coresets. Aside from this, obtaining average distortion bounds is an ill-posed problem in both theory and practise. Without any measure over the candidate solutions, defining the average distortion is impossible in theory. Certainly, a uniform measure over all solutions is meaningless as any candidate solution has density 0. In experiments, one could easily generate a large number of candidate solutions where the distortion is guaranteed to be small and articificially (and meaninglessly) decrease the average distortion. For example if the centers are sufficiently far away from a good constant factor approximation, the coreset will always have low distortion.

---

> ### Author Response · Authors · 2022-07-28
> **$\ell_z$ Clustering**
>
> The reason why the analysis cannot be extended to arbitrary powers of $z$ is fairly in-depth in the analysis. Given that we could not add all the details for questions that we can prove within the page limit, we seriously question why we should use space on questions that we could not prove.
>
> Furthermore, it is not easy to explain why things go wrong without a lot of familiarity, or without many explanations taken on faith. There are two instances where the analysis will fall short. We will focus on the more easily understood one. Here, the short answer is that even in constant dimension, it is not known how to obtain a coreset of size $k\varepsilon^{-2}$ for arbitrary powers of $z$ (whereas Group Sampling achieves this for k-median and k-means). Where this appears in the analysis is not easy to see.
>
> For a slightly more indepth answer, hopefully the following comments will help.
> Any analysis of the cost estimator that results in a linear dependence on k has a variance of $\varepsilon^{-z}$ (both for Group Sampling and Sensitivity Sampling where this bound is even higher). This bound also appears in Lemma 3 where it is one part of the minimum over which we take trade off the values $k_i$ and $2^i$. Naively, this leads to a dependency of at least $k \cdot d\cdot \varepsilon^{-2} \cdot 2^i$, where $d$ is the dimension. $2^i$ can be bounded by $\varepsilon^{-z}$ for powers of z, using additional arguments. Thus, at this point, we have to find a way to remove $d$ and $\varepsilon^{-z}$ from this bound.
>
> We remove $d$ via the chaining analysis. For the initial anchoring solution in the chaining argument (i.e. the first step in the chaining analysis), we require a coreset of size $k \cdot \varepsilon^{-2} \cdot d'$, where $d'$ is the dimension of the space containing the anchoring solutions. In particular $d'$ is not the same thing as the initial dimension of $d$. In this paper, we show that $d'$ can be chosen to be $\sqrt{k}$. Unfortunately, this still leaves a dependency of $k \cdot \sqrt{k} \cdot \varepsilon^{-2} \cdot 2^i \leq k^{1.5}\cdot \varepsilon^{-2} \cdot \varepsilon^{-z}.$ The dependency on $\varepsilon^{-z}$ can be reduced to $\max(\varepsilon^{-z+2},1)$, yielding a coreset of size $k^{1.5} \varepsilon^{-z}$. How this final step can be achieved technically is a fairly involved algebraic manipulation. Details appear on pages 20-22 in the supplementary pdf.
>
> Since it is already known that Group Sampling yields coresets of size $\tilde{O}(k \varepsilon^{-z})$, the bound above yields no improvement. If the reviewer would like further details, we will provide them.

---

> ### Comment · Reviewer_1Yz9 · 2022-08-08
> **Score change following rebuttal**
>
> I thank the authors for the detailed response. I have gone through the paper once again with the new details and explanations in mind.
> I will change my score as all my questions have been answered.

---

### Official Review · Reviewer_jVkM · 2022-07-10

**Rating:** 6
**Confidence:** 4
**Soundness:** 3 good
**Presentation:** 2 fair
**Contribution:** 3 good

**Summary:**

The paper presents a theoretical analysis leading to smaller coreset sizes for the problem of $k$-means. The coreset can then be extended also to the $k$-median problem.

The authors provide an improvement over the work of Cohen-Addad et al. [2022], by finding nets (discretizes set of solutions) with better properties, to obtain their coreset size. This in term leads to the smallest coreset size for $k$-means with the emphasis that the coreset size should not be linearly in $k$.

**Questions:**

1) What is $p$ at the inequality below Eq. 2? I assume $p \in C_i$, right?
2) Why does it imply that after $t = \frac{1}{\varepsilon^{-2}}$ rounds, $(I - \Pi_U\Pi_U^T)s = \overset{\to}{0}$? please clarify this point.
3) From what I understood, $Pi_U$ is the span of all previously added vectors from Lemma 5. So basically, this is a set, right? If so, what does it mean to have $\left\| Pi_Us - s^\prime \right\|$? Please do clarify.

**Limitations:**

The authors clearly addressed the limitation of their work.

**Strengths And Weaknesses:**

* Strengths:
    1) A smaller coreset size for the k-means problem, leading to a subquadratic coreset size in $k$ (the number of means).
    2)  While the paper is based on a paper by Cohen-Addad et al. [2022], the authors present techniques that hinge upon finding better nets for obtaining smaller coreset sizes for both the $k$-means and $k$-median problems. In other words, the authors present an improvement on the techniques of Cohen-Addad et al. [2022], by generalizing the techniques to rather finding nets with better properties.

* Weaknesses:
The paper is somewhat hard to follow, and also contains many grammatical errors.

---

> ### Author Response · Authors · 2022-07-28
> **Reply to Reviewer 3**
>
> 1. $p$ is indeed in $C_j$
> 2. To expand on the proof of Lemma 5: If we set $t> 1/\varepsilon^2$ then the inequality in the third to last line becomes at least 1. Since the inequality is also always at most 1, as it is the absolute value of the inner product between two unit vectors, it means that it is equal to 1.
> By construction, all of the vectors $(I-\Pi_{U_{i}}\Pi_{U_{i}}^T)p$ lie in the span of $U$ and moreover, these vectors are are pairwise orthogonal. Thus, $s$ can be written as linear combination of these vectors and the projection of $s$ onto the kernel of the space spanned by $U$, which is $(I-\Pi_U \Pi  _U^T)s$ must be the origin.
> 3. $\Pi_U$ is the orthogonal matrix whose rows span all the vectors contained in the subspace spanned by $U$. $\Pi_Us$ is the projection of $s$ onto that subspace.

---

> > ### Comment · Reviewer_jVkM · 2022-08-08
> > **Raising my score**
> >
> > I thank the authors for the clarifications. In light of this, my score has been raised.

---

### Official Review · Reviewer_22jd · 2022-07-11

**Rating:** 5
**Confidence:** 3
**Soundness:** 3 good
**Presentation:** 3 good
**Contribution:** 3 good

**Summary:**

The paper studies coresets for k-means clustering in d-dimensional Euclidean space. The paper is of theoretical nature and shows an improved upper bound of $O^\ast(\epsilon^{-2}k^{1.5})$ on the size of coresets that can be obtained by group sampling. This improves the current best bound of $O^\ast(\epsilon^{-2}k^2 \min\{\epsilon^{-2},k^2\})$.

**Questions:**

My suggestion is to improve the technical parts. Also the plots of the experiments may be improved (one can barely see the differences in cost.)

**Limitations:**

No potential negative societal impact in sight.

**Strengths And Weaknesses:**

The paper studies an important and interesting problem, which is under active research. The obtained upper bound is a further step towards closing the gap to the current lower bound.

The work is somewhat incremental since it heavily uses results from previous work and basically achieves the result by improving a part of it -- the cost vector net.

On the other hand, the obtained bound is nice and is certainly worth to be published.

Another weakness is that the notation and the technical details are not always clean and convincing. Also the experiments are not that meaningful.

---

> ### Author Response · Authors · 2022-07-28
> **Reply to Reviewer 2**
>
> We respectfully disagree that the improvement is incremental. This is the first result that improves over the quadratic dependency on k while retaining an optimal dependency in k. No previous analysis in the entire coreset literature managed to do this. Aside from this, it also gives strong evidence that a future analysis of Group Sampling will show optimality (up to $polylog(k/\varepsilon)$ factors).
>
> In terms of the analysis, it is known that the bound on the variance of the coreset estimator is tight. Therefore, the only challenge towards obtaining an optimal bound for coresets will be an improvement for the nets. The fact we improve the analysis of an existing estimator / coreset algorithm should not be taken as a lack of novelty, in our opinion, as if the Group Sampling algorithm is optimal, it is unclear why research should focus on a different algorithm.
>
> If there are specific questions regarding notation and technical details, we will be happy to answer them.
>
> We would also appreciate details regarding the comment that the experiments are not meaningful.

---

> > ### Comment · Reviewer_22jd · 2022-08-08
> > **Reply to the Authors**
> >
> > Dear authors,
> >
> > the work is indeed a bit incremental since it is based on ideas from previous works -- and improves a part of these. The result is very nice though, as I have already stated in my review. However, compared to a submission that introduces completely new ideas and directions, this is a small weakness. Nevertheless, I think that your work is worth publishing!
> >
> > To the experiments: First, the cost plots need a scaling. As we are dealing with very large values there, one can barely observe differences between the different algorithms. (Maybe try a logarithmic scale?) Second, the sample size does only depend on $k$ and it is not completely clear what a distortion of e.g. $1.05$ means. Is that an empirical $epsilon$ of $0.05$? It would be nice if the experiments would also be conducted with respect to different $\epsilon$ and then instead of a distortion an empirical $\epsilon$ is plotted.

---

> > > ### Author Response · Authors · 2022-08-08
> > > **Reply to Reviewer 2**
> > >
> > > Distortion of 1.05 means an $\varepsilon$ of 0.05.
> > > The difference in distortion is clearly visible, even at this scale. The difference in cost is indeed negligible and hard to see, but in our view and for the purpose of this paper also unimportant. The cost only provides a qualitative information that indeed, computing coresets is different from optimizing on these data sets. If this were not the case, one should not evaluate the data set to begin with, as it becomes easy to compute a coreset, as long as the cost substantially drops with increasing $k$.
> > >
> > > It is not clear what you mean by conducting the experiments with respect to different $\varepsilon$.
> > > There are three ways to conduct these experiments: Fix $k$, fix $\varepsilon$, or vary both. Varying both at the same time would be possible, but it is very hard to tell which parameter made a difference. Fixing $k$ and varying $\varepsilon$ only allows us to empirically measure the constants, but does not give any further insights. For this reason, we settled for a fixed $\varepsilon$ with varying $k$. This evaluation can at least give a hint as to what the correct dependency on $k$ is, which is also the main topic of this paper. Note that we are working under the assumption that the correct dependency on $\varepsilon$ is $\varepsilon^{-2}$.
> > >
> > > Please note that the experiments are only supposed to support our theoretical results. We did not aim to provide a general experimental evaluation of coresets since this question is out of scope of this paper and has been studied previously in other works for a number of algorithms, including those presented here.

---

> > > > ### Comment · Reviewer_22jd · 2022-08-09
> > > > **Reply to the Authors**
> > > >
> > > > So the cost plots could as well be completely removed, right? Then one could conduct experiments for varying $(\epsilon, k) \in \{ 0.25, 0.5, 0.75 \} \times \{ 10, 20, \dots, 50 \} $ for example.

---

> > > > > ### Author Response · Authors · 2022-08-09
> > > > > **Reply to Reviewer**
> > > > >
> > > > > There is still value in those plots. The only thing that is unimportant is the variance in those costs. But we can remove them, if you so insist. We have other plots for the theoretical hard instance in the supplementary material with which we could replace the cost plots.
> > > > >
> > > > > As mentioned before, varying $\varepsilon$ does little other than determining constants. Varying $k$ and $\varepsilon$ together is very difficult to interpret, it is not clear when varying both parameters which causes an effect, and it is not clear what additional information we could gain from them other than what our plots already show.

---

### Official Review · Reviewer_XcEV · 2022-07-24

**Rating:** 6
**Confidence:** 3
**Soundness:** 4 excellent
**Presentation:** 3 good
**Contribution:** 4 excellent

**Summary:**


The paper considers the problem of providing a bound on the size of the coresets for the K-means problem which is closer the lower bound than the existing state of the art approaches. Some experimental results are also provided to support the analysis in the paper.


**Questions:**

(A)Does this analysis follow through for other clustering problems such K-median, K-mediods etc?

(B) It seems like you are able to get a better clustering nets because you can bound the error by the geometric mean (smaller) than the arthimetic mean. Is this the tightest one can get?

(C) The experimental section is showing the cost of the various algorithms as well as the distortion results. However, it would be interesting to show the scaling of the size of the coresets. These are provided in the supplementary and would be better served if shown in the main paper?


**Limitations:**

The paper would be a nice contribution to the clustering literature but I wonder if it would be better suited to a primarily theoretical audience like COLT or similar. I provided an initial lower rating to reflect that and will adjust it based on the upcoming discussions.


**Strengths And Weaknesses:**

The paper is highly compressed to fit into the confines of the conference submission. Though the writing is clear, it heavily relies on reading up on the results from Cohen-Addad et al's recent papers.

The setting is of prime interest given that K-means shows up in various machine learning settings. The result of improving the min(k, 1/eps^2) to the \sqrt(k)
is pretty significant because now it is clearer that there will be a potential algorithm matching the lower bound of O(k/eps^2).

I checked over the main theorems and approach but did not work through all the details.

The paper is solving an important problem in clustering but the reduction from k to \sqrt(k), while theoretically very interesting, is a bit too narrow.

---

> ### Author Response · Authors · 2022-07-28
> **Reply to Reviewer 1**
>
> The analysis also works for Euclidean k-median (with some modifications), and we gave details how the analysis changes in the supplementary material. The bound on the coreset size still remains $O(k^{1.5}\varepsilon^{-2})$. As far as we are aware, the k-mediod objective is simply k-median with the centers being drawn from input points. As such a k-median coreset also implies a k-mediod coreset.
>
> The bound $k^{1.5}$ is the best one can hope for using Lemma 3. The analysis does not really use the geometric mean, unless you are considering an interpolation between $k^{1.5}\varepsilon^{-2}$ and $k\varepsilon^{-4}$. If we have $k_i = 2^i = \sqrt{k} \ll \varepsilon^{-2}$ then the two tradeoff bounds $k \varepsilon^{-2} \left(\frac{k\cdot k_i\cdot 2^i}{(k+k_i\cdot 2^i)^2}\right)\cdot \min(k_i,2^i)$ agree at $k^{1.5} \cdot \varepsilon^{-2}$.
>
> For the experiments: We can do our best to fit in more such details in the main body for the final version.

---

### Public Comment · ~Lingxiao_Huang2 · 2022-11-23
**Regarding the proofs of the paper ''Improved coresets for the Euclidean $k$-means problem''**

Dear Authors,

We read the paper and have major difficulties understanding many places in the proofs. Some issues seem to be quite serious and require nontrivial effort to fix. We hope to get clarifications from the authors.


## Major issues
- **Lemma 5**. This is a key lemma for bounding the net size in Lemma 4. However, the statement of the lemma is incorrect.
$\quad$ When the authors apply Lemma 5 to prove Lemma 4, it seems that they need the following claim instead of Lemma 5: for any $\alpha > 0$, it is possible to find $|U| = O(\alpha^{-2})$ such that $$\forall p\in P, |p^T (I-\Pi_{U}\Pi_{U}^T) s|\leq \alpha\cdot \|(I-\Pi_{U}\Pi_{U}^T)p\|\cdot \mathrm{min}\_{q\in P} \|q-s\|.$$  We need to show the above inequality for all $\alpha>0$, not the specific $\varepsilon$ (which is the precision required for the coreset). This is a minor problem.
More seriously, one can easily verify that the statement of the lemma is incorrect. For instance, letting $s=p$ for some point $p\in P$, we have $\mathrm{min}\_{q\in P} \|q-s\| = 0$, which implies that the right side is 0. Consequently, for any $p\in P$, we conclude that the left side should be 0, which implies that $U$ spans the whole dataset $P$. However, the dimension of $\mathrm{span}(P)$ can be $d \gg O(\alpha^{-2})$. This is a contradiction with the conclusion that $|U| = O(\alpha^{-2})$.
- **Ineq. (10)**. This is a key for bounding the sample number $|\Omega|$. It applies Lemma 8 in the first inequality. However, it seems that the use of Lemma 8 is incorrect.
$\quad$ The authors claim that the cardinality $|N\_{h+1}\times N\_h|$ is at most $\mathrm{exp}(\gamma\cdot k\cdot \log\|P\|\_0\cdot \mathrm{min}(k_i+2^{2h},2^{2h}\cdot 2^i)\cdot i\cdot h)$ due to Lemma 4. However, Lemma 4 only considers a fixed sub-collection $\mathcal{A}$ with $k_i$ clusters. For bounding the net size, it seems that we should consider all possible $\mathcal{A}$'s together with different $k_i$'s. Consequently, the cardinality $|N_{h+1}\times N\_h|$ should be $$\sum\_{k\_i,\mathcal{A}: |\mathcal{A}| = k\_i} \mathrm{exp}(\gamma\cdot k\cdot \log\|P\|\_0\cdot \mathrm{min}(k_i+2^{2h},2^{2h}\cdot 2^i)\cdot i\cdot h).$$
More importantly, by Lemma 8, Ineq. (10) could only use the variance upper bound for all $C$, i.e., we can only use the term $\max\_{k\_i} \frac{k\cdot k\_i\cdot 2^i}{(k+k\_i\cdot 2^i)^2}$ instead of $\frac{k\cdot k_i\cdot 2^i}{(k+k_i\cdot 2^i)^2}$ in the variance upper bound. However, by this modification, it seems that we can not get the desired sample number for $|\Omega|$ since $\max\_{k_i} \frac{k\cdot k_i\cdot 2^i}{(k+k_i\cdot 2^i)^2} = \Theta(1)$ and $\mathrm{min}(k_i, 2^i)$ can be large up to $\mathrm{min}(k,\varepsilon^{-1})$.

## Minor issues
- **Proof of Lemma 4**. The proof is very difficult to read. It seems that there are several typos and missing details.
$\quad$ The exponential sequence $\alpha^2(1+\alpha/2^i)^j$ is confusing. Should it be $(1+\alpha/2^i)^j \alpha \sqrt{\mathrm{cost}(p,\mathcal{A})}/2^i$? Also, should the range of $j$ be $O(\alpha^{-1} \cdot 2^i\cdot i)$?
$\quad$ The authors do not show how to construct cost vectors clearly. For every $s$ on the net, should we use $(\|\Pi_U (p-s)\|^2)\_{p\in P}$ as cost vectors? Or should we use $(\|\Pi_U (p-s)\|^2 + \|(I-\Pi_U \Pi_U^T)p\|^2)\_{p\in P}$ as cost vectors? Or something else?
- **The inequality before (6)**. Symbol $\mathcal{S}$ still appears in the denominator $|\Omega|\cdot (\mathrm{cost}(P,\mathcal{S}) + \mathrm{cost}(P,\mathcal{A}))$, but disappears in the subscript of $\mathrm{sup}$ which only contains the enumeration of net points.
- **Notations in Lemma 4**. Both $\mathcal{C}$ and $\mathcal{A}$ are reused symbols.


Best,
Lingxiao, Jian, Xuan

---

> ### Public Comment · ~Chris_Schwiegelshohn1 · 2022-11-23
> **Reply to the major issues.**
>
> Hello Lingxiao, Jian, and Xuan,
>
> regarding Lemma 5, if there is indeed a point where $s=p$, then this point is unique. In this case, the rank of the subspace spanning $U$ would be 1 and not $d$ and the process stops after a single round. If you have a very sparse linear combination such that $s$ is written as, say, a linear combination of 100 points, then the rank would be 100. In case this is the confusion: your example would mean that $s$ is orthogonal to the nullspace of $U$ and thus $(I-\Pi_U\Pi_U^T)s=0$.
>
> As for the application of Lemma 8: the number of all subcollections of clusters of size $k_i$ is indeed ${k \choose k_i} \leq 2^k$. Multiplying the net by sizes by $2^k$ does not increase $\log |N_{h}|$ by more than a constant factor.
> EDIT:
> Perhaps I should say, we enumerate over all subsets, each of which has a net of size $t$. Then we get nets of size $2^k \cdot t$. Now $t$ happens to be of the order $exp(k\cdot 2^{-2h}\min(k_i,2^i)\cdot O(1))$, if we disregard polylogs. Thus the increase in net size by enumerating over all subsets is absorbed by the $O(1)$.
>
> Finally, our arxiv version only makes a claim for k-median ($z=1$) and k-means ($z=2$). For the remaining values of $z$, we do not claim anything.
>
> Best,
> Chris

---

> > ### Public Comment · ~Lingxiao_Huang2 · 2022-11-24
> > **Reply to the NeurIPS version**
> >
> > Thanks a lot for the clarifications!
> >
> > Regarding Lemma 5, now we understand your statement is correct. We guess it might be better to change the use of the Pythagorean theorem as follows:
> > $$1\geq \sum_{i\in [t]}\big(\frac{p_i^T(I-\Pi_{U_{i-1}} \Pi_{U_{i-1}}^T)(I-U_0 U_0^T)s}{\|(I-\Pi_{U_{i-1}} \Pi_{U_{i-1}}^T)p_i\|\cdot\|(I-U_0 U_0^T)s\|}\big)^2\geq t\varepsilon^{-2},$$ where $p_i$ is the point selected in round $i$.
> >
> > Regarding the application of Lemma 8, you may miss the question about the variance upper bound. We guess that one possibility to fix the issue is to divide $k_i$ into $\log k$ intervals $[2^i,2^{i+1}-1]$. For each interval, we can get Ineq. (10). The overall error is at most the sum of errors for each interval, which increases at most $\log k$ times.
> >
> > Other than the above minor point about the variance, we have no additional questions. We will properly cite your paper in our arXiv paper soon
> > (we already updated the arXiv version and it should appear in a few days).
> >
> > btw: dear Chris, you misspelled our names.
> >
> > best,
> > Lingxiao, Jian and Xuan

---

> > > ### Public Comment · ~Chris_Schwiegelshohn1 · 2022-11-24
> > > **Reply to the NeurIPS version**
> > >
> > > Fixed your names in the above comment. Sorry about that.
> > >
> > > As for the arxiv version, we are currently convinced of our claims and see no reason to retract them. Eventually, we will upload a new version to account for your and other comments. Please contact us via email about that paper, if you have further questions.
> > >
> > > Best, Chris

---

### Public Comment · ~Lingxiao_Huang2 · 2022-11-23
**Other issues in the arXiv version (https://arxiv.org/abs/2211.08184)**

Dear authors,

We also found that the authors submitted an arXiv version very recently.
The arXiv version is an extension of the NeurIPS version. In the arXiv version, the authors claimed new bounds for general $(k,z)$-clustering. In fact, in the NeurIPS version, the authors claimed that the technique does not extend to the general $z$ case. But the arXiv version essentially used the same technique.
After reading the arXiv version, we found several computations for this generalization are not correct. We hope to get clarification.

- **The end of Page 10**. The authors claim that $\mathrm{min}\_{p\in P} \|p-s\|\leq O(1)\cdot 2^{i/z}\cdot \mathrm{cost}(p,\mathcal{A})^{1/z}$. Why does the index term $1/z$ of $\mathrm{cost}(p,\mathcal{A})$ disappear in the bound for $\|p^\top (I-\Pi_U \Pi_U^\top)s\|$, which is concluded from the bound of $\mathrm{min}\_{p\in P} \|p-s\|$?
- **The last paragraph on Page 13**. The authors claim that
$$\mathrm{min}(k_i,2^i) \frac{k\cdot k_i \cdot 2^{i(z-1)}}{(k+k_i 2^i)^2}\in O(k^{z/(z+2)}),$$ which is the key for bounding the coreset size. However, this claim is obviously incorrect. For instance, by letting $z=1$, we have that the maximum value of the left side is 1 instead of $O(k^{z/(z+2)}) = O(k^{1/3})$ by letting $k_i = 2^i = k^{1/2}$.

Finally, we would like to bring to the authors' attention our recent concurrent work (arXiv: 2211.11923). The work was completed a few months ago and is currently under submission. One result we obtained in our paper is the same coreset size bound $O(k^{(2z+2)/(z+2)})$ for general $(k,z)$-clustering in the Euclidean case (we have some optimal/improved results for other metrics). The high level idea is somewhat similar (we also try to tighten the variance using a different error term), but the details are quite different. We noticed your work a few days ago and hence decided to upload our paper to arXiv immediately. We also tried to cite your results properly as a concurrent work (either the $k$-means result or the general $z$ results). But we have had major difficulty in understanding the proofs. We think several places are incorrect and fixing them is not straightforward (of course we believe everything is ``eventually fixable" since the same bound is already proved in our paper). So we are currently uncertain about how to cite the paper properly and will really appreciate the clarifications from the authors (and we will be happy to revise the citation to the paper in our arXiv paper if the main issues mentioned above can be clarified).

Best,
Lingxiao, Jian, Xuan

---

> ### Public Comment · ~Chris_Schwiegelshohn1 · 2022-11-23
> **reply to the bound on k-median**
>
> You may be correct. Your calculation is not (for your choice of parameters it is $O(1)$), but there may be a calculation issue. We will check and update the arxiv version as soon as possible if it is indeed the case. Other than that, contact us personally for issues with the arxiv paper. We will not discuss anything on OpenReview that does not pertain to the NeurIPS version.

---

> > ### Public Comment · ~Lingxiao_Huang2 · 2022-11-25
> > **Reply to the arXiv version**
> >
> > Thanks a lot for the reply! We will contact you via email for issues with the arXiv version as suggested.

---

### Meta-Review · Area_Chair_mSij · 2022-08-24

**Recommendation:** Accept
**Confidence:** Less certain

**Metareview:**

I actually really liked the paper despite the weak accept reviews - and decided to bump this up. Improving the known coreset bound for k-means is extremely important, as that is a fundamental problem. While it does indeed heavily build upon prior work and doesn't get optimal bounds, it is giving a novel analysis for a very important problem (with a huge line of prior work with worse bounds), and thus I recommend acceptance.

**Award:**

No

---

### Decision · Program_Chairs · 2022-09-14

Accept